# Can machine-learning algorithms improve upon classical palaeoenvironmental reconstruction models?

Peng Sun[1], Philip. B. Holden[2], and H. John B. Birks[3,4]

[1]Institute of Environmental Sciences (CML), Leiden University, 2333 CC Leiden, the Netherlands
[2]Environment, Earth and Ecosystem Sciences, The Open University, Walton Hall, Milton Keynes, MK7 6AA, UK
[3]Department of Biological Sciences and Bjerknes Centre for Climate Research, University of Bergen, PO Box 7803, Bergen N-5020, Norway [4]Environmental Change Research Centre, University College London, London WC1 6BT, UK

*Correspondence to*: Philip B. Holden (philip.holden@open.ac.uk)

**Abstract.** Classical palaeoenvironmental reconstruction models often incorporate biological ideas and commonly assume that the taxa comprising a fossil assemblage exhibit unimodal response functions of the environmental variable of interest. In contrast, machine-learning approaches do not rely upon any biological assumptions, but instead need training with large data-sets to extract some understanding of the relationships between biological assemblages and their environment. To explore the relative merits of these two approaches, we have developed a two-layered machine-learning reconstruction model MEMLM
(Multi Ensemble Machine Learning Model). The first layer applies three different ensemble machine-learning models (random forests, extra random trees, and lightGBM), trained on the modern taxon assemblage and associated environmental data to make reconstructions based on the three different models, while the second layer uses multiple linear regression to integrate these three reconstructions into a consensus reconstruction. We considered three versions of the model: 1) a standard version of MEMLM, which uses only taxon abundance data, 2) MEMLMe, which uses only dimensionally reduced assemblage
information, using a natural language-processing model (GloVe) to detect associations between taxa across the training data-set, and 3) MEMLMc which incorporates both raw taxon abundance and dimensionally reduced summary (GloVe) data. We trained these MEMLM model variants with three high quality diatom and pollen training sets and compared their reconstruction performance with three weighted averaging (WA) approaches (WA-Cla, classical deshrinking; WA-Inv, inverse deshrinking; and WA-PLS, partial least squares). In general, the MEMLM approaches, even when trained on only dimensionally reduced
assemblage data, performed substantially better than the WA approaches in the larger trainng sets, as judged by cross-validatory prediction error. When applied to fossil data, MEMLM variants sometimes generated qualitatively different palaeoenvironmental reconstructions from each other and from reconstructions based on WA approaches. We applied a statistical significance test to all the reconstructions. This successfully identified each incidence where the reconstruction is not robust with respect to the model choice. We found that machine-learning approaches could outperform classical
approaches, but could sometimes fail badly in the reconstruction, despite showing high performance under cross-validation, likely indicating problems when extrapolation occurs. We found that the classical approaches are generally more robust, although they could also generate reconstructions which have modest statistical significance, and therefore may be unreliable. Given these conclusions, we consider that cross-validation is not a sufficient measure of transfer-function performance, and

we recommend that the results of statistical significance tests are provided alongside the down-core reconstructions based on fossil assemblages.

## 1 INTRODUCTION

The distribution and abundance of taxa are interrelated with the environment (Ovaskainen et al., 2017). By considering environmental variability across space instead of through time, the palaeoenvironment can be reconstructed by applying modern taxon-environment relationships to the fossil record (e.g. Battarbee et al., 2005; Cleator et al., 2020; Turner et al., 2020).

With the development of palaeoecological research, large training data-sets for environmental reconstruction have been compiled in recent years. (e.g. Harrison, 2019a; Bush et al., 2021). Data assimilation has long been a focus of Earth science and ecology, and the integration of larger data-sets provides more comprehensive training information (e.g. Christin et al., 2019; de la Houssaye et al., 2019; Bush et al., 2021). For large data-sets, machine-learning methods have strong advantages and may be appropriate to extract the non-linear relationships between taxon compositional information and the environment, and to integrate a variety of sources of data (e.g. Helama et al., 2009; Aguirre-Gutierrez et al., 2021; Wei et al., 2021b).

In recent years, machine learning has been applied to a wide range of applications in palaeoecology (Hais et al., 2015; Jordan et al., 2016). Wei et al. (2021b) reconstructed palaeoclimate using five different machine-learning methods based on digital leaf physiognomic data and integrated the predictions by averaging. Hais et al. (2015) predicted the Pleistocene biota distributions in palaeoclimate using machine learning. Huang et al. (2020) used one series of palaeoclimate sequences to predict the climate in another period. These studies show that machine learning has strong versatility and effectiveness, and suggest it could be more widely applied.

Machine-learning approaches are not based upon any biological assumptions, which may weaken their performance relative to mathematically simpler classical approaches that do. For instance, weighted averaging (WA) approaches are based upon the simple but realistic assumption that taxa have a unimodal response to the environmental variable of interest (ter Braak and Barendregt, 1986). The absence of any such prior understanding is likely to place additional demands on the minimum adequate size of a modern training set. Moreover, it may weaken the ability of machine learning to operate under extrapolation, critically important when applying any reconstruction approach to past taxon assemblages that lack modern analogues. To address these questions, we have developed the Multi Ensemble Machine Learning Model (MEMLM) to apply in a systematic comparison with classical WA reconstruction approaches.

The benefit of machine learning lies in its robust data mining and information extraction capabilities, especially when applied to large data-sets. Data mining involves discovering patterns, trends, and correlations hidden within extensive data-sets. Information extraction, on the other hand, focuses on extracting insights from unstructured data, typically relying on Natural Language Processing and encoding techniques to understand and analyse relationships within unstructured data. An associated

problem is that when a sample size is limited, machine learning is more likely to learn the noise component and generate prediction errors due to over-fitting (Yeom et al., 2018; Syam and Kaul, 2021). This suggests that an ensemble-learning method, which integrates models with potentially different biases, may improve the prediction performance (Wei et al., 2021b). Ensemble learning was developed to address these issues (Zhou, 2012) and is the motivation for the ensemble learning approach we present, namely the Multi Ensemble Machine Learning Model (MEMLM).

We build MEMLM from three different machine-learning ensemble models of random forests, extra random trees, and lightGBM. We then combine these three models into a single consensus model which we treat as our 'best' machine-learning approach. Classical studies have integrated different ecological approaches by calculating the mean of their predictions (Norberg et al., 2019). An arithmetic mean gives equal weight to each model, even though the models may have different advantages in different applications (Schulte and Hinckley, 1985; Zhou, 2012). In MEMLM, we weight each model according to its predictive power under cross-validation.

Most classical models give equal weight to different taxa, which may reduce their prediction potential and smooth the reconstruction (e.g. Brooks and Birks, 2001; Heiri et al., 2003; Battarbee et al., 2005; Wei et al., 2021a). In WA-PLS (TWA-PLS), tolerance down-weighting can be applied to assign weights to each taxon in reconstructing the environment that depends upon the breadth of the taxon's environmental niche (Liu et al., 2020). Bayesian approaches such as BUMPER (Holden et al., 2017) are built on classical assumptions and are highly constrained by taxa with low environmental tolerances, especially when characterised with high confidence. In machine-learning ensemble models, each taxon has a different predicted contribution which is used to weight its contribution to the ensemble.

We develop three versions of MEMLM; the standard version which only considers raw taxon abundance data; MEMLMe, which only uses dimensionally reduced assemblage data; and MEMLMc, which uses both. The motivation for the dimensional reduction is to explore whether considering known associations between taxa can improve the palaeoenvironmental reconstructions. For this, we use the natural language processing model GloVe (Pennington et al., 2014), which calculates the relationships between co-occurring words in the same sentence. GloVe is a form of dimension reduction which assigns vectors (also called embedding) to each word according to the word connection relationships, so that each sentence can be represented as a superposition of the word embeddings within that sentence. In taxon assemblages, there are analogous co-occurrence relationships between taxa which we hypothesise convey information on their ecological functioning. We therefore use GloVe to generate embedding vectors by considering the frequency of co-occurring taxon pairs across the training set. We then concatenate the embedding vectors of each sample to represent the assemblage.

In summary, there are several aspects to the question of whether machine-learning algorithms can improve upon classical reconstruction methods. Our strategy to address these has three components

1) There are many ensemble machine-learning algorithms, and there is no reason to prefer any of these a priori. To address this, we apply three widely used approaches of random forests, extra random trees, and lightGBM. We combine these into a single consensus reconstruction to simplify comparisons and provide the 'best possible' reconstruction.

2) Natural language-processing models are a widely used dimensional reduction approaches in machine learning, and we apply one such method, GloVE, to supplement ensemble machine learning trained on raw count data. We explore whether this approach can usefully encode assemblage information to either i) improve the reconstructions based only on raw count data - unlikely given that dimension reduction does not provide additional information, but not ruling out the possibility that data transformation can assist the learning or ii) replace the raw count data, increasing numerical efficiency and potentially providing information on ecological functioning.

3) It is not sufficient that a reconstruction approach performs well on a training set. It must also be statistically robust when applied to independent core data, which likely lies outside the high-dimensional space of the training set. We cannot assume that machine learning and classical approaches perform equally well under extrapolation. Therefore, we do not only apply conventional tests of cross-validated RMSEP, regression slope and $R^2$, derived solely from the training set, but we also consider the statistical significance of core reconstructions, applying the technique of Telford and Birks (2011)

## 2 MATERIALS AND METHODS

We apply MEMLM to high quality pollen and diatom training sets to generate down-core reconstructions. We calculate training set cross-validation metrics and we quantify the statistical significance and robustness of the core reconstructions. We compare these performance metrics with those of classical WA approaches to evaluate whether, and under what circumstances, machine-learning approaches might be able to outperform classical WA-based reconstruction approaches.

### 2.1. MEMLM

MEMLM combines a series of modules (Figure 1). In this section, we introduce the functions of each module and the data processing approach. There are three model variants (MEMLM, MEMLMe, and MEMLMc), each of which takes different inputs (Figure 1), which is the only difference in their construction. The scientific motivation for the three variants is to explore i) whether machine-learning decision trees can extract all useful information (MEMLM), or, if not, ii) whether GloVe can improve this (MEMLMc) and iii) whether GloVe alone is sufficient to encode assemblage data (MEMLMe). Each variant is built using the same three machine-learning approaches (random forests, extra random trees, and lightGBM), which are combined into a single consensus reconstruction model for each.

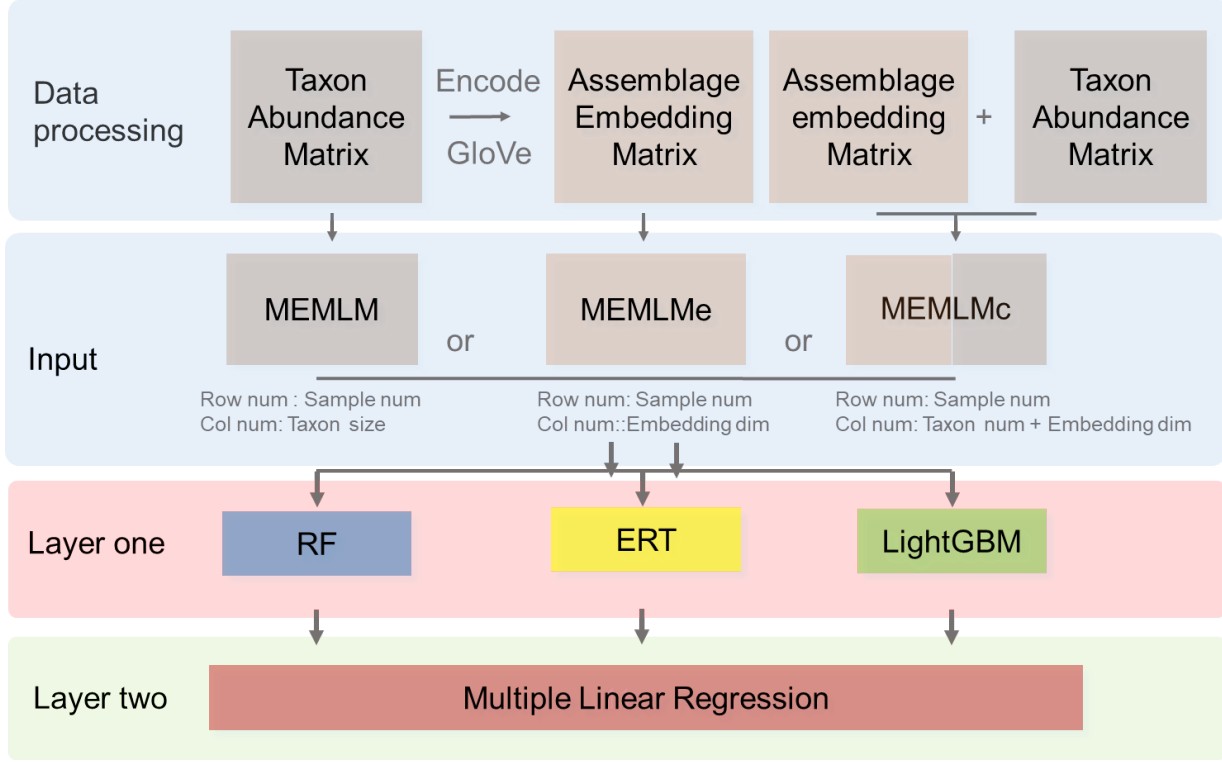

**Figure 1: Multi Ensemble Machine Learning Model (MEMLM) model framework. MEMLM has a modular building block architecture so that components can be easily changed. Raw num and Col num are the number of rows and columns in the input matrix; dim is the number of dimensions.**

### 2.1.1. First layer

The input data comprise environmental data together with either the taxon abundance matrix, the assemblage embedding matrix, or both matrices (see section 2.2.3 for a description of the embedding algorithm).

We apply three ensemble machine-learning models to derive the mapping between taxon composition information and environmental factors:

(1) Random forests (RF) is an ensemble machine-learning model composed of multiple decision trees. The overall model framework is determined based on the predictive power of each decision tree applied to the training data-set under bootstrapping. Individual decision trees with better predictive performance are allocated higher weights, and the 'forest' integrates the weighted result from each tree (Liaw and Wiener, 2002).

(2) Extra Random Tree (ERT) is similar to RF, except that it uses the entire data-set rather than a bootstrapped subset (Geurts et al., 2006).

(3) LightGBM is based on the Gradient Boosting Decision Tree. This also integrates decision trees, but LightGBM differs by applying 'gradient boosting' to add new trees, building each new model on the residuals of the previous model to improve the prediction. It has the ability to merge sparse data-sets to increase computational efficiency (Friedman, 2001; Ke et al., 2017).

**2.1.2 Second layer (consensus reconstruction)**

It is possible to improve prediction performance by integrating the prediction of multiple models into a consensus reconstruction (Yeom et al., 2018; Syam and Kaul, 2021). Averaging is widely used to integrate the output prediction of multiple models. However, the integration weight of each model is the same under averaging. MEMLM applies multiple linear regression to allocate an integration weight to each model rather than attaching each model with the same weight. The

150 consensus reconstruction is derived as follows. The three upstream models are applied to reconstruct the training data-set and we then build a multiple linear regression model to fit the reconstructed values to the actual value in the training set. To fit the multiple linear regression model, we apply internal 5-fold cross-validation for each model separately and use the predictions from this cross-validation to fit regression weights. We then treat the consensus model as a single encapsulated model and perform 5-fold cross-validation, each time using 80% of the training set. The total validation computation therefore comprises

155 five internal cross-validations and one regression fit. This approach is designed to avoid the risk of over-fitting while reducing the impact of low-performance models on the consensus reconstruction. In an exploratory analysis applied to the NIMBIOS data-set, building models for each of 18 environment attributes demonstrated that the multiple linear regression approach reduced the root mean square error of prediction (RMSEP) relative to the individual reconstructions by an average of 8% (Table A1). A consensus reconstruction based on the mean of the three ensemble approaches also improved predictive power

but reduced the cross-validated RMSEP errors relative to the individual reconstructions by an average of 5%. We note that while the consensus approach reduces RMSEP by typically 8%, we show in Section 3.1 and Table 1 that such improvements are modest relative to the improvements from the machine learning itself. Weights of the linear models of MEMLM, MEMLMe, and MEMLMc based on the three training sets are provided in Table A2.

**2.1.3 Embedding**

The GloVe algorithm (Pennington et al., 2014) is a very widely used linguistic dimensional reduction approach. It uses co-occurrences of words in phrases to characterise numerically their meaning. In formal terms, GloVe is a row-column bilinear model of the form $r_i + c_k + R_i \times C_j$, fitted by weighted least-squares to the log-transformed co-occurrence matrix derived from the primary data. GloVe is thereby very close to unconstrained ordination models used in ecology except perhaps for the

170 transformation to co-occurrences (ter Braak, 1988, ter Braak and te Beest, 2022). GloVe is trained on assemblages to map taxa onto vectors in feature space, so that the assemblages can be described as linear combinations of the features.

It may be helpful to describe the motivation for this particular row-column model. In GloVe, words are represented as vectors in high dimensional space, where each dimension captures an aspect of meaning so that in this space words that have similar meanings are located near to each other. To illustrate, in word vector space, we would expect the difference vectors $\boldsymbol{queen} - \boldsymbol{king}$ and $\boldsymbol{girl} - \boldsymbol{boy}$ to be similar, as they both reflect only a change of gender, with other dimensions of meaning (species, age, social status etc) constant. Embedding reduces the dimensionality of a vocabulary from tens of thousands of words to hundreds of similar meaning dimensions, known as features.

In ecology, co-existence among taxa can reflect characteristics of the environment (Ovaskainen et al., 2017). We hypothesise that taxa within an assemblage have relationships that are analogous to words within a phrase, so that in the feature space of ecological 'meaning' the vectorial representation of a taxon describes its ecological function. We apply GloVe to ecological assemblages. Instead of analysing co-occurrences of words within phrases, we analyse co-occurrences of taxa within assemblages. The objective is to extract ecological information by associating taxa with their ecosystem functioning.

The GloVe algorithm is fully detailed in Pennington et al. (2014), and here we introduce the underlying philosophy and illustrate it in the context of ecological functioning. Consider $P_{ij}$ the conditional probability that taxon j appears in the same assemblage as taxon i:

$$P_{ij} = P(j|i) = X_{ij}/X_i \tag{1}$$

where $X_{ij}$ is the number of assemblages which contain both taxa i and j, and $X_i$ is the number of assemblages containing taxon i. This probability does not necessarily indicate the strength of the relationship. Consider, for instance, that a high value may simply reflect that taxon j is common and therefore provides little information about the environment.

To determine associative relationships, GloVe considers the ratio $P_{ik}/P_{jk}$ where taxon k is some probe taxon used to differentiate the ecological functioning of i and j. If taxon k has a strong association with taxon i but not with taxon j then $P_{ik}/P_{jk} \gg 1$. However, if all three taxa are either commonly found together or have no relationship (i.e. low but random co-occurrence) between each other, $P_{ik}/P_{jk} \sim 1$, indicating that taxon k provides very little information to help distinguish the ecological functions of i and j. The value of $P_{ik}/P_{jk}$ can therefore inform us about the direction of difference vector $i - j$. For application to MEMLMc, the feature matrices are provided together with the raw taxon count data to provide richer training data for the ensemble-learning algorithms.

## 2.2. Assemblage data

For model training purposes we use two large pollen data-sets, SMPDSV1 (Harrison, 2019a) and NIMBIOS (Bush et al., 2021), and the smaller diatom SWAP data-set (Stevenson et al., 1991). The SMPDSV1 (Harrison, 2019) and SWAP (Stevenson et al., 1991) data-sets record the percentage of each taxon in each sample, whereas the NIMBIOS data-set uses integer counts. When constructing the co-occurrence matrix, whether the data are integer counts or percentages, we sum that data during co-occurrence. To demonstrate the palaeoenvironment reconstructions of each model, we apply i) SWAP to reconstruct lake-

water pH from diatoms in a core from The Round Loch of Glenhead (RLGH) (Allott et al., 1992, Jones et al., 1989), ii) SMPDSV1 to reconstruct mean temperature of the coldest month (MTCO) from pollen in the Villarquemado core (Harrison, 2019a, Harrison, 2019b), and iii) NIMBIOS to reconstruct the mean annual temperature (MAT) from pollen in the Consuelo (Urrego et al., 2010) and Llaviucu (Steinitz-Kannan et al., 1983, Colinvaux et al., 1988) cores.

### 2.2.1. Training data-sets

SWAP: The SWAP training set (Stevenson et al., 1991) was developed as part of an international scientific effort directed at establishing and understanding the impacts of acid rain on freshwaters. It includes relative abundance data for 277 diatom taxa from 167 modern samples with clear identification criteria standards (Birks et al., 1990). We apply these data to reconstruct lake-water pH.

The NIMBIOS data-set (Bush et al. 2020), includes samples from 636 neotropical locations with various habitat types. There are 533 pollen types (some taxa can only be identified to family level), ranging from soil samples to mud-water interface samples from lakes. We use it to reconstruct mean annual temperature (MAT).

The SMPDSv1 data-set was developed as an environmental calibration data-set to provide training data for palaeoclimate reconstructions (Harrison, 2019a). SMPDSv1 contains the relative abundancies of the 247 most important pollen taxa in 6458 terrestrial samples from Europe, northern Africa, the Middle East, and Eurasia, compiled from multiple different published sources. We use it to reconstruct mean temperature of the coldest month (MTCO).

### 2.2.2. Core data-sets

We apply the SWAP training set to the RLGH and RLGH3 core data-sets. RLGH is a fossil diatom data-set from The Round Loch of Glenhead, Scotland, taken to explore anthropogenic acidification (Allott et al., 1992). The data-set includes the relative abundances of 41 diatom taxa in 20 samples which span the industrial era. RLGH3 was sampled to explore natural acidification driven by weathering and soil development during the Holocene (Jones et al., 1989). This data-set includes abundances for 225 diatom taxa in 101 samples.

We apply the NIMBIOS training set to the Consuelo and Llaviucu core data-sets. The core from Lake Consuelo, Bolivia, is an 8.8 m sediment sequence, which records the long-term evolution of cloud forest in response to environmental changes over the last 46,300 years (Urrego et al., 2010). Lake Llaviucu is a temperature-sensitive lake in the Ecuadorian Andes (Steinitz-Kannan et al., 1983; Colinvaux et al., 1988). It lies behind a moraine in the system dated by Clapperton (1987) within the last glaciation (35,000 yr B.P.). At nearly 37 degrees S latitude, the lake is perched on the eastern face of the Cordillera Occidental and has been lifted 2,200 m since deglaciation. It shows the possibility of significant cooling of tropical latitude rain-forest near San Juan Bosco (Colinvaux et al., 1997).

We apply the SMPDSv1 training set to the Villarquemado core data-set (Harrison 2019, Wei et al 2021a), a pollen record from the western Mediterranean Basin spanning the interval from the last part of MIS-6 to the late Holocene. The fossil pollen data

were assigned to the subset of pollen taxa recognised in the modern SMPDSv1 data-set. There are 104 taxa represented in the final taxon list based on the 361 core samples.

## 2.3. Model parameters, performance, and validation metrics

### 2.3.1. Model parameters

We build the GloVe model using the PyTorch deep learning frame (Paszke et al., 2019), which provides a set of tools and interfaces to implement, train, and deploy deep-learning models. In embedding training, we set the number of epochs (training loops) to 1,000 and the number of embedding dimensions to 256. For the first layer, we build an ensemble of 1,000 decision trees with parallel computing. MEMLM has an external interface so that these parameters can be easily changed for any third-party application.

We originally developed the GloVe analysis using the pre-packaged software 'glove-python' [https://github.com/maciejkula/glove-python] but subsequently re-wrote the GloVe algorithm from first principles. Cross-validation and down-core reconstructions from the two algorithms were not materially different and so the statistical significance testing, which is highly expensive computationally, requiring one month of parallel computing, was not repeated.

### 2.3.2. The prediction importance indicator for taxon weighting

The MEMLM models are ensembles based on the results of multiple decision trees. Each time a decision tree forks, the algorithm explores different ways to integrate each taxon's abundance to increase predictive power. The algorithm works through an internal cross-validation analysis to determine whether each predictor reduces the prediction errors in each decision tree, and then summarises the results across all decision trees. The approach ascribes an importance index to each taxon which is normalised to a total of 1 across all taxa and provides a measure of that taxon's predictive power. The ten most important taxa for each of the three machine-learning models are listed in Table A3. These are used in the inference of taxon importance for environmental reconstruction.

### 2.3.3. Uncertainty quantification

Uncertainty quantification is provided for all machine-learning reconstructions using IBM's UQ360 package (IBM 2024). UQ360 utilizes meta-models to estimate the uncertainty bounds of the preserved models, providing upper and lower limits on prediction errors. Specifically, it employs additional decision-tree models to capture and re-estimate the prediction errors of the source models.

### 2.3.4. Cross-validation

The predictive powers of the MEMLM variants are compared with classical WA models (ter Braak and Barendregt, 1986) and WA-PLS (ter Braak and Juggins, 1993). We take RMSEP, regression slope, and $R^2$ score as performance evaluation indicators,

using the scikit-learn package (Pedregosa et al., 2011). We use five-fold cross-validation. We perform each cross-validation five times with random shuffling allowing us to provide mean estimates for all validation metrics along with their standard deviations, which we provide for RMSEP. We note that spatial correlation and pseudo-replication within a training set can lead to overstated cross-validated performance statistics. These problems can be minimised by, for instance, removing sites that are geographically close and environmentally similar (Liu et al., 2020). However, we include all training-set sites in cross-validation, noting that our objective is to compare the relative performances of different approaches applied to the same training sets.

For evaluation of the classical models we use the rioja package in R (Juggins, 2017) with default settings. As WA-PLS performance is sensitive to the number of components; we accept a higher PLS component only if it exhibits a 5% improvement in RMSEP on the previous component (Birks, 1998) and we present results for the higher component.

### 2.3.5. Statistical significance of reconstructions

While cross-validation is a useful measure of predictive power which implicitly guards against over-fitting (Yates at al., 2023), it is likely to over-estimate predictive power in practice as fossil assemblages may lie outside the high dimensional space of the modern training assemblages, for instance by lacking close modern analogues. Telford and Birks (2011) developed an easily applied method for testing the robustness of a reconstruction of a specific sequence. The approach is to create an ensemble of transfer functions using the same biological assemblage as the training set, but with randomised values of the environmental variable, and calculating the proporion of variance in the fossil data explained by a single reconstruction. If the reconstructed variable is found to explain more of the variance than 95% of the random reconstructions, then the reconstruction is deemed to be statistically significant. We apply this approach with the palaeoSig package in R (Telford and Trachsel, 2015) to all core reconstructions as an indicator of their robustness.

### 2.4. Computing hardware

In this study, the computing CPU is Intel Core i7-4710MQ; the model is supported by the scikit-learn package (Pedregosa et al., 2011), a powerful machine-learning Python package which incorporates the most widely used machine-learning algorithms and related data processing and validation functions. MEMLM supports parallel computing: with more CPU cores, the computing time will decrease significantly. The computational time taken for five-fold cross-validation of the MEMLMc model is 138 seconds (SWAP), 406 seconds (NIMBIOS), and 2834 seconds (SMPDsV1).

### 3 RESULTS

### 3.1. Cross-validation

Table 1 compares the cross-validated RMSEP for the three training sets and the six reconstruction approaches (see Figure A1 for regression visualization of predicted values against observed values). Regression slope and $R^2$ score are also provided. All

validation data are the means of five separate cross-validation exercises, which are also used to provide a percentage error estimate for RMSEP (in brackets). WA-PLS is found to be the best performing classical approach in all three training sets as

evaluated by RMSEP, but in each case it is outperformed by MEMLM, which reduces RMSEP by 6% (SWAP, 167 training samples, 277 taxa), 22% (NIMBIOS, 636 training samples, 533 taxa), and 50% (SMPDSv1, 6548 samples, 257 taxa). The benefits of machine-learning approaches clearly increase with increasing training-set size.

MEMLMe is trained only on embedded assemblage data from GloVe. The approach does not work well for the SWAP training set, but it significantly improves upon WA approaches when using the larger NIMBIOS and SMPDVs1 training sets,

suggesting that when the training set is large enough, embedding is able to extract most of the predictive power of the assemblages. However, MEMLMe is consistently the worst performing MEMLM variant (albeit generally better than the WA approaches), and so we do not use it in the reconstructions.

We performed additional cross-validation tests on MEMLMe to confirm that the embedding approach can encode useful information, noting that with an embedding dimension of 256 (comparable to the number of taxa in the training sets) we are

not applying the approach under significant dimensional reduction. To explore this, we applied a range of embedding dimensions to the MEMLMe model of the richest data-set, namely the 533-taxon NIMBIOS data-set (Figure A2a). This sensitivity analysis demonstrates that 30 dimensions are sufficient for MEMLMe to outperform WA-PLS (RMSEP 2.914°C) in this training set. Figure A2b illustrates the learning power of increased training, with RMSEP increasing by around 0.4°C as the number of training epochs is reduced from the 1,000 we used to 40.

MEMLMc uses both the taxon abundance and the embedding matrices. These additional data do not significantly affect the predictive performance relative to MEMLM under cross-validation, suggesting that conventional ensemble machine-learning approaches are sufficient to encode adequately the assemblage information in training sets comprising a few hundred taxa. However, we retain this model for down-core reconstructions to explore whether the addition of embedding information can affect reconstructions in a way that is not captured by RMSEP.

|  |  | MEMLM | MEMLMe | MEMLMc | WA-Inv | WA-Cla | WA-PLS(best) |
|---|---|---|---|---|---|---|---|
| **RMSEP** |  |  |  |  |  |  |  |
| SWAP | pH | **0.290 (3.7%)** | 0.331 (3.1%) | 0.296 (2.8%) | 0.308 (1.1%) | 0.317 (1.0%) | 0.308 (1.1%) |
| NIMBIOS | MAT/ °C | 2.254 (1.6%) | 2.221 (1.2%) | **2.094 (1.4%)** | 3.176 (0.5%) | 3.587 (0.6%) | 2.923 (0.6%) |
| SMPDSv1 | MTCO/ °C | **2.353 (0.5%)** | 2.779 (0.9%) | 2.478 (0.6%) | 5.310 (0.1%) | 6.672 (0.1%) | 4.979 (0.2%) |
| Slope |  |  |  |  |  |  |  |
| SWAP | pH | 0.984 | 1.002 | 0.999 | 1.029 | 0.899 | 1.030 |
| NIMBIOS | MAT/ °C | 0.996 | 0.998 | 0.999 | 1.005 | 0.750 | 0.996 |
| SMPDSv1 | MTCO/ °C | 0.997 | 0.997 | 0.997 | 1.000 | 0.629 | 0.996 |

R$^2$ score

| | | | | | | |
|---|---|---|---|---|---|---|
| SWAP | pH | **0.858** | 0.815 | 0.852 | 0.840 | 0.831 | 0.840 |
| NIMBIOS | MAT/ °C | 0.856 | 0.860 | **0.876** | 0.714 | 0.635 | 0.758 |
| SMPDSv1 | MTCO/ °C | **0.926** | 0.897 | 0.918 | 0.624 | 0.407 | 0.670 |

MAT mean annual temperature; MTCO mean temperature of the coldest month

Table 1. Cross-validated root mean square error of prediction (RMSEP), regression slope, and R$^2$ score for the three training sets. All data are the means of five cross-validation exercises, which are also used to provide uncertainty estimates for RMSEP (error for RMSEP in brackets, expressed as a percentage of RMSEP). MEMLM uses the abundance matrix. MEMLMe uses the assemblage embedding matrix. MEMLMc uses the combined abundance and embedding matrices.  WA-Cla is weighted averaging with a classical deshrinking regression, WA-Inv is weighted averaging with an inverse deshrinking regession (Birks et al., 1990). WA-PLS is the 'best' model (see section 2.2.3), see Table A4 for other components. Bold highlights the model with the lowest RMSEP or highest R$^2$ score.

### 3.2. Environmental reconstructions and comparisons

For each core we compare the reconstructions from the models with lowest RMSEP, being the MEMLM and MEMLMc machine-learning approaches and the best classical approach (section 2.3.3), which is WA-PLS using one component for SWAP and WA-PLS using two components for NIMBIOS and SMPDSV1. In the Appendix, Figures A3 to A7 illustrate scatterplot matrices of all six reconstruction approaches, and Figures A8 to A12 compare reconstructions for all six models through time. In each reconstruction we additionally provide the statistical significance test results (Telford and Birks, 2011). A reconstruction is considered significant when that reconstruction explains more of the variance than 95% of 1,000 randomised reconstructions, based on the same training assemblage but with randomised environmental values.

### 3.2.1. pH reconstructions from RLGH using the SWAP training set

MEMLM and WA-PLS1 show similar trends of acidification, with pH declining from around 5.2 at about 1870 to around 4.8 at about 1980 (see Figure 2). MEMLMc shows a similar trend but with reduced acidification relative to the other approaches. All three reconstructions are statistically significant, and with high explained variance, though WA-PLS1 explains more variance (58%) than MEMLM (46%) or MEMLMc (52%.). The variance explained by the first principal component of the fossil core assemblages is 62%, indicating that the reconstructed pH explains most of the dominant part of the variance in the fossil diatom assemblages.

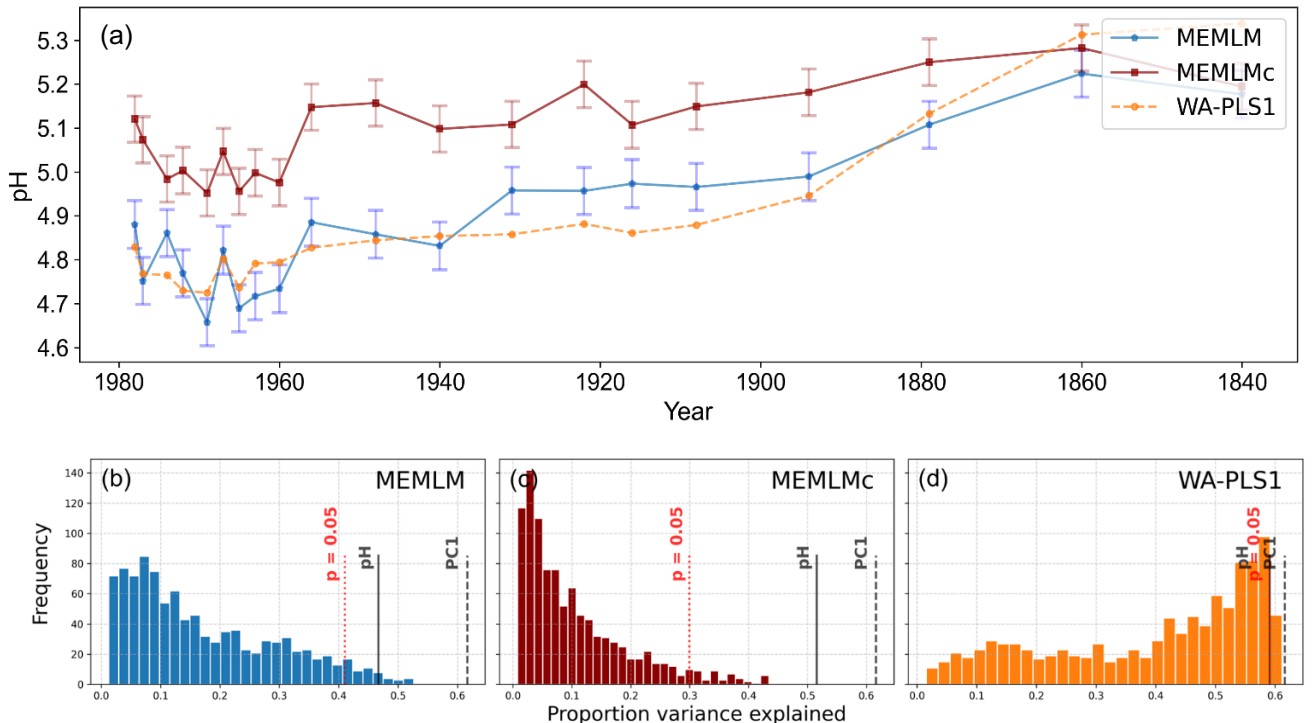

Figure 2: a) pH reconstruction for the RLGH core. b, c & d) statistical significance testing of MEMLM, MEMLMc, and WA-PLS1 reconstructions, respectively. MEMLM uncertainties are calculated using IBM UQ360 (section 2.3.3). These compare with cross-validated RMSEP errors of 0.292 (MEMLM), 0.294 (MEMLMc), and 0.308 (WA-PLS1) pH units.

### 3.2.2. pH reconstruction from RLGH3 using the SWAP training set

All three methods provide reconstructions that show similar trends of lake-water pH, with gradual acidification in the early record from around 5.6 to 5.2 pH, attributed to the development of organic soils (Jones et al., 1989) and then a rapid post-industrial acidification from around 5.2. to 4.8 pH. The three reconstructions also exhibit similar variability, previously attributed to loss of tree cover and peat erosion (Jones et al., 1989), further suggesting reconstruction robustness. Moreover, all three reconstructions are statistically significant, explaining between 23% and 27% of the core variance, which compares to 32% variance explained by the first principal component of the fossil assemblages (Figure 3).

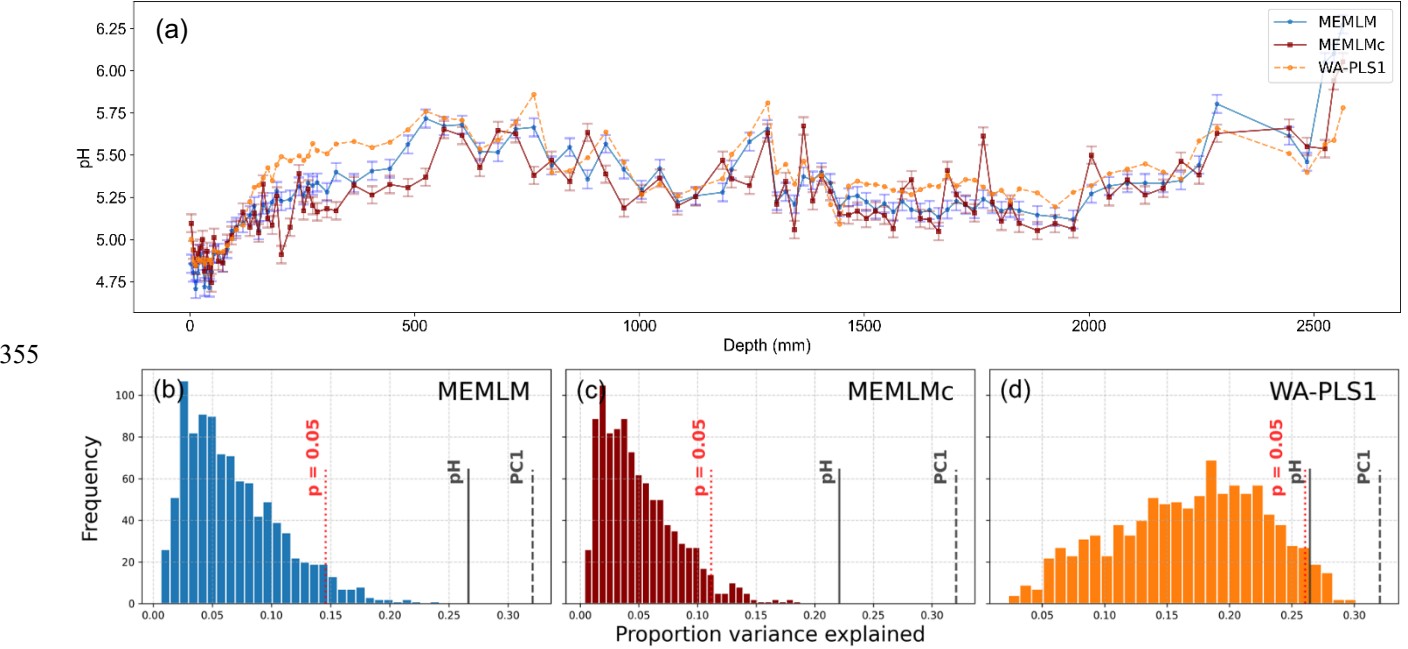

**Figure 3: a) pH reconstruction for the RLGH3 core. b, c & d) statistical significance testing of MEMLM, MEMLMc, and WA-PLS1 reconstructions, respectively. MEMLM uncertainties are calculated using IBM UQ360 (section 2.3.3). These compare with cross-**
360 **validated RMSEP errors of 0.292 (MEMLM), 0.294 (MEMLMc), and 0.308 (WA-PLS1) pH units.**

### 3.2.3. MAT reconstruction from Consuelo using the NIMBIOS training set

All three methods display similar trends, most notably reconstructing about a 4°C warming from the Last Glacial Maximum at 21,000 BP to the start of the Holocene at 11,000 BP. The MEMLM approaches are more variable in general, although variability is largely synchronous between the three reconstruction approaches and may be associated with Dansgaard-
365 Oeschger (D/O) events (Bond et al., 1993; Blunier and Brook, 2001). At 8000 BP, WA-PLS2 displays a 10°C cooling excursion which is not apparent in the MEMLM reconstructions. Although a cooling event at 8.2ka is well known, the cooling reconstructed by WA-PLS2 seems excessive. All three methods are statistically significant and explain core assemblage variance of between 27% and 29%, compared to 32% explained by the first principal component (Figure 4).

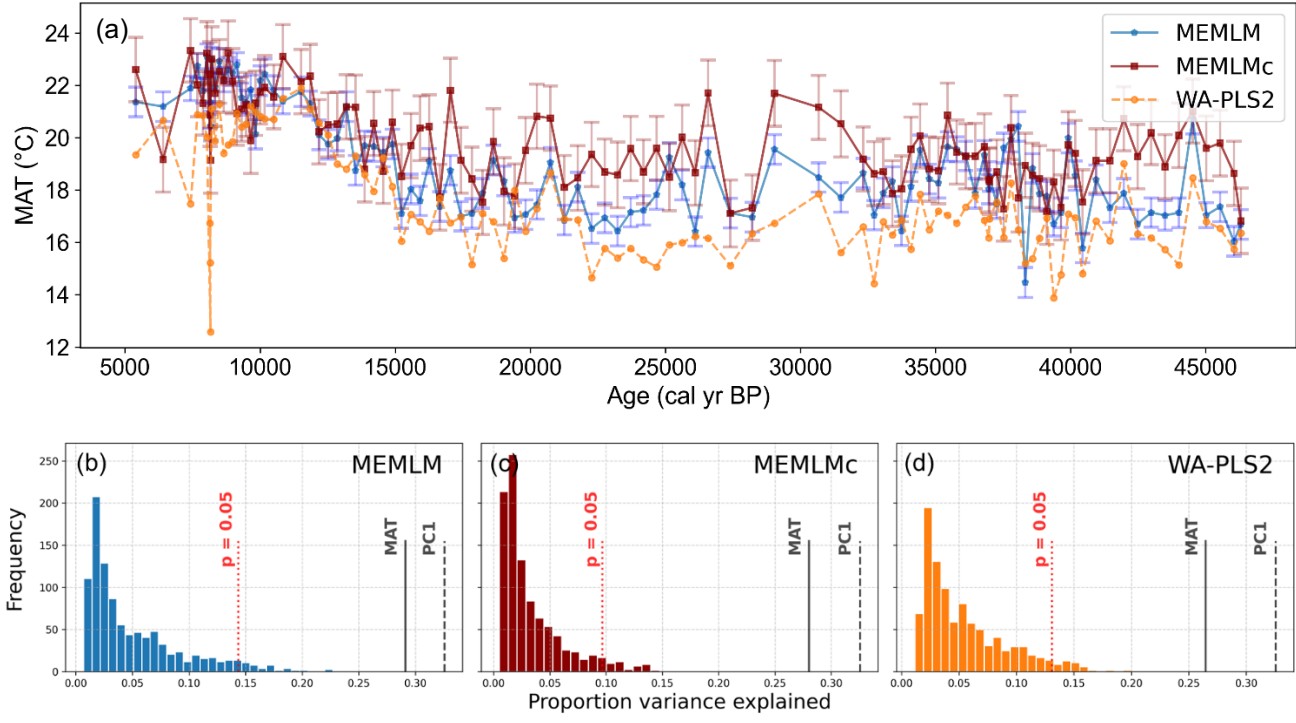

**Figure 4: a) MAT reconstruction for the Consuelo core. b, c & d) statistical significance testing of MEMLM, MEMLMc, and WA.PLS2 reconstructions, respectively. MEMLM uncertainties are calculated using IBM UQ360 (section 2.3.3). These compare**
**with cross-validated RMSEP errors of 2.254 (MEMLM), 2.094 (MEMLMc), and 4.979 (WA-PLS2) °C.**

### 3.2.4. MAT reconstruction from Llaviucu using the NIMBIOS training set

All three methods display similar overall trends with mid-Holocene warming, but each display different centennial variability, which for the MEMLMc reconstruction is clearly unrealistic for the Holocene, with temperature excursions as large as 8°C. Neither of the MEMLM approaches are statistically significant at the 95% confidence level, so neither can be accepted as
robust. The WA-PLS2 reconstruction is statistically significant, although it only explains 13% of the core-assemblage variance compared to the 28% explained by the first principal component of the core data (Figure 5).

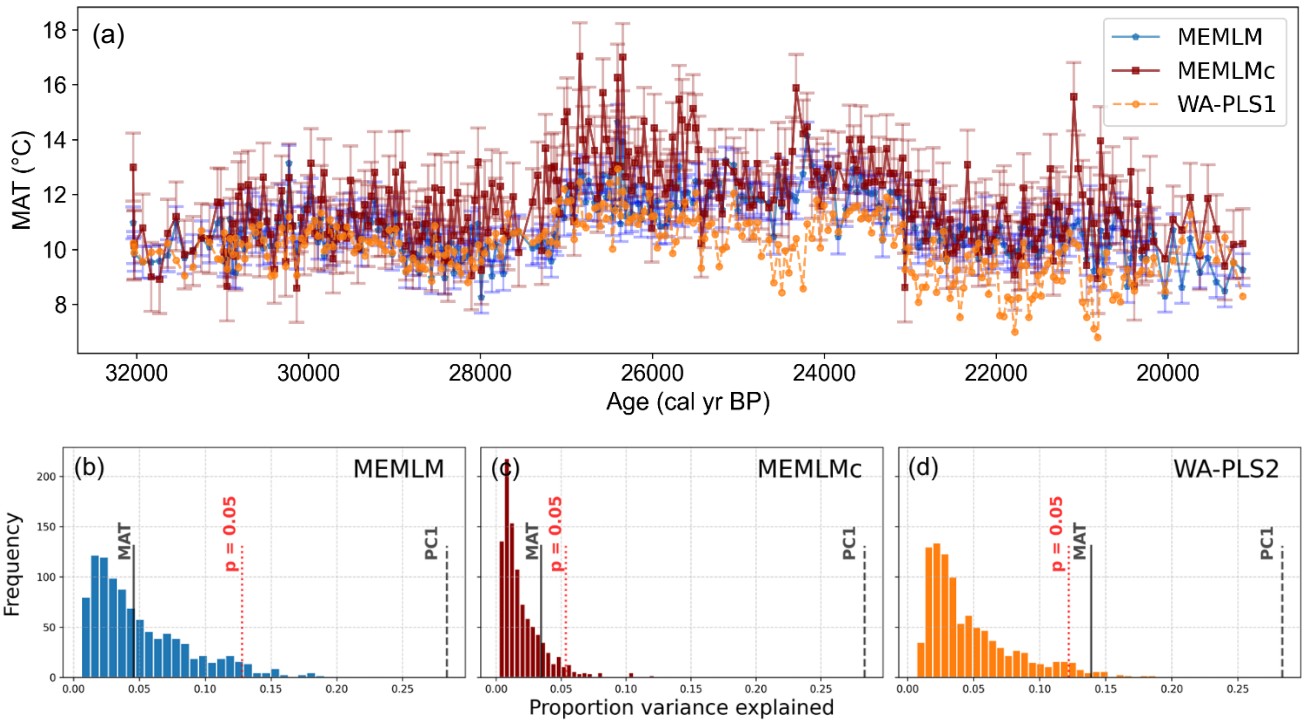

Figure 5: a) MAT reconstruction for the Llaviucu core. b, c & d) statistical significance testing of MEMLM, MEMLMc, and WA-PLS2 reconstructions, respectively. MEMLM uncertainties are calculated using IBM UQ360 (section 2.3.3). These compare with cross-validated RMSEP errors of 2.254 (MEMLM), 2.094 (MEMLMc), and 4.979 (WA-PLS2) °C.

### 3.2.6. MTCO reconstruction from Villarquemado using the SMPDSV1 training set

All three approaches generate noisy reconstructions with high variability that is incoherent. It is difficult to discern any meaningful trends. None of the reconstructions, including WA-PLS2, are statistically significant. The low (17%) variance associated with the first principal component suggests that the fossil assemblages are responding to multiple environmental factors with responses that are too complex to be captured by a single explanatory environmental variable (Figure 6).

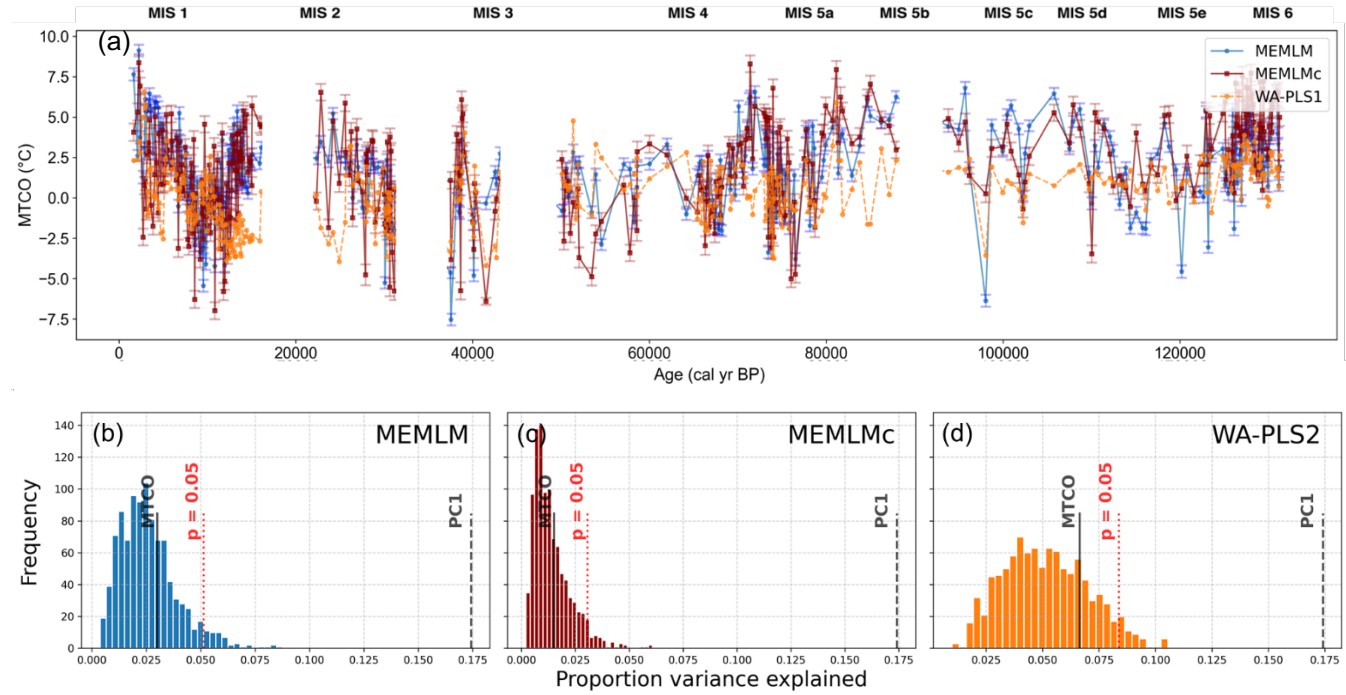

**Figure 6: a) MTCO reconstruction for the Villarquemado core. b, c & d) statistical significance testing of MEMLM, MEMLMc, and WA-PLS2 reconstructions, respectively. MEMLM uncertainties are calculated using IBM UQ360 (section 2.3.3). These compare with cross-validated RMSEP errors of 2.353 (MEMLM), 2.434 (MEMLMc), and 2.923 (WA-PLS2) °C.**

## 4 Discussion and conclusions

We have developed three variants of a multi-model ensemble machine-learning algorithm, MEMLM. These each train three separate ensemble machine-learning algorithms (random forests, extremely random trees, and lightGBM) and combine them into a consensus reconstruction using multiple regression. The three approaches only differ in their input data. The simpler MEMLM takes only taxon abundance data. MEMLMe, built only upon the GloVe embedding matrix, does not perform as well as MEMLM. However, MEMLMe was found to be a useful reconstruction model, at least when applied to the larger NIMBIOS and SMPDSV1 training sets, and the embedding usefully summarises taxon assemblages with fewer than 50 dimensions. Our motivation for retaining 256 embedding dimensions in MEMLMe is that the focus of GloVe is on extracting semantic meaning. In linguistics, typically 200 dimensions of meaning are needed to encode fully a language. While we have shown that far fewer dimensions are sufficient to build a good reconstruction model, demonstrating the explanatory power of the most important embedding dimensions, there are progressive improvements in performance as dimensional size increases (Fig. A2). This demonstrates that less important dimensions can provide useful explanatory information, and potentially additional understanding and interpretability.

The additional complexity of MEMLMc, which uses both taxon count and embedding, did not significantly affect the predictive performance relative to MEMLM under cross-validation, suggesting that conventional ensemble machine-learning approaches are sufficient to encode adequately ecological information in the relatively small data-sets used in these palaeoclimate reconstructions. We note that the real power of embedding (dimension reduction) approaches is likely to be in their applications to much larger data-sets, when ecological relationships between 10,000s of taxa and their environment are being considered.

We have focussed only on a comparison with weighted averaging approaches, which are the most widely used reconstruction technique, being simple to apply, well understood, and straightforward to interpret. The MEMLM approaches are found to perform better than classical weighted averaging approaches under cross-validation. In the case of the smallest SWAP data-set the advantages are modest, but in the largest SMPDSV1 data-set RMSEP errors are reduced by a factor of two relative to the best performing classical WA approach. These improvements in performance clearly validate the potential benefits of strong data-mining abilities of machine learning, suggesting these techniques have the potential to improve upon classical reconstruction approaches.

When applied to core reconstructions, MEMLM approaches generate considerably more variability than the WA-PLS reconstructions. While some elements of this additional variability might be realistic, especially considering that WA-PLS approaches are known to bias reconstructions towards the centre of their training data (Liu et al., 2020), the variability is not always coherent between different reconstruction approaches and the magnitude of MEMLM variability is in some cases implausibly high, for example by suggesting Holocene variability of up to 8°C in the Ecuadorian Llaviucu core.

We performed significance testing on all core reconstructions and found that five of the fifteen reconstructions are not statistically significant and therefore are not considered robust. Both MEMLM and MEMLMc approaches fail on the Llavuicu core, confirming our suspicion that the unrealistic variability is an artefact. All three approaches fail the statistical robustness test at Villarquemado, which is sensitive to multiple environmental factors and has responses which appear too complex to be captured by a single explanatory variable.

The shapes of the histograms of the proportion of variance explained in the RLGH and RLGH3 pH reconstructions based on diatom data and randomised modern SWAP training pH values in the significance testing are very different for WA-PLS1 and for MEMLM and MEMLMc (Figs. 2, 3). Such differences contrast with the more consistent histogram shape for the significance-test results for the other sequences where the reconstructions are based on pollen data (Figs. 4–6). Machine-learning approaches generally fail badly when trained with randomised environmental data as the histograms are left-skewed and explain little down-core variance (Figs. 2–6). In contrast, the WA-PLS1 pH reconstructions (Figs. 2, 3) based on diatom data explain a substantial amount of the down-core variance even when the modern pH data are randomised (Figs. 2, 3). This may result from the short and dominant environmental gradient in the SWAP diatom–pH training data and the high inherent correlation and dominance of a relatively few abundant taxa within the modern and fossil diatom data. The pollen training data, however, used for the MAT or MTCO reconstructions of the other sequences (Figs. 4–6) are large (638 and 6,458

samples) and hence cover longer and more complex environmental gradients than the pH training data (167 samples). It is also likely that the pollen data, both modern and fossil, are influenced by multiple environmental factors, not only MAT or MTCO. In summary, while MEMLM can generate useful reconstructions, it should always be used in conjunction with statistical significance testing to confirm that the reconstructions are robust and potentially realistic and reliable. The additional

450 complexities of incorporating embedding information in MEMLMc does not reduce RMSEP or spurious variability and neither does it improve statistical significance. However, MEMLMe demonstrates that embedding is useful as it can summarise ecological assemblages using significantly fewer dimensions. Its benefits may be clearer in applications with much larger data-sets and in applications beyond palaeoenvironmental reconstructions. The poor performance of MEMLM in some reconstructions may be due to extrapolation due to no-analogue fossil assemblages. All models are applied under the same

extrapolation. The WA-PLS2 reconstructions exhibit higher statistical significance than MEMLM, although WA-PLS2 also fails to generate robust reconstructions at Villarquemado. We infer that that the use of simpler WA models, which include a major biological assumption (unimodal environmental response) can be more powerful than the use of brute-force learning, despite reductions in RMSEP. We reiterate our recommendation that all reconstructions using any approach, should be accompanied with statistical significance testing. Seemingly useful models may fail when applied under extrapolation or when

the assemblage variance is only weakly dependent on the reconstructed environmental variable.

**Acknowledgements**

PS was funded by a PhD scholarship from the China Scholarship Council (CSC, no. 202104910033). HJBB's participation has been possible thanks to the European Research Council under the European Union's Horizon 2020 Research and Innovation Programme grant agreement 74143 to the project 'HOPE: Humans on Planet Earth - long-term impacts on biosphere dynamics'

awarded to HJBB. We thank Mark Bush and Alex Correa-Metrio for generously providing the NIMBIOS modern pollen data and Llaviucu fossil pollen data and Graciela Gil Romera for generously providing the SMPDS v1 modern pollen data and Villarquemado fossil pollen data. We are grateful to Cajo ter Braak and Andrew Parnell for valuable comments that improved the manuscript. HJBB thanks Cathy Jenks for her invaluable help.

**Data availability**

All data-sets can be found in the cited data-sets and articles in the references, except the RLGH3 and Llaviucu core-data. The latter were made available to us by Mark Bush.

**Code availability**

All codes are available in github.   WA and WA-PLS use the *rioja* package (https://github.com/nsj3/rioja).  Telford and Birks (2011) statistical significance uses *randomTF* in the *palaeoSig* package (https://github.com/richardjtelford/palaeoSig).

MEMLM is available at https://zenodo.org/records/13138593

**Competing interests**

The authors declare that they have no conflict of interest.

**Author contributions**

PS conceptualised the application of GloVe to ecological assemblages. PS, PBH, and HJBB conceptualised the experimental design. PS developed the MEMLM model and performed all analyses and graphical visualisations. HJBB contributed assemblage data. PS and PBH wrote the manuscript with reviewing and editing by HJBB.

## Appendices A

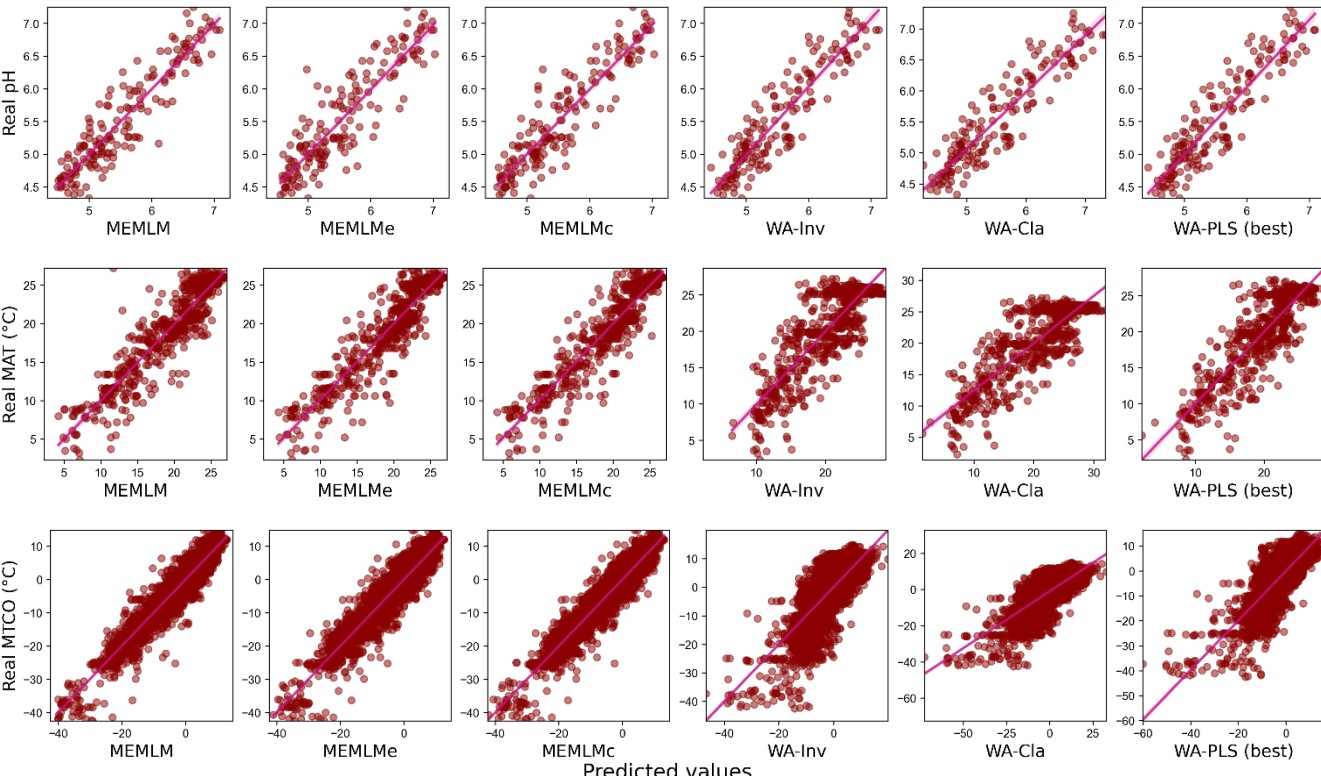

**Figure A1: Scatterplots of observed values against predicted values in three training sets. MEMLM uses the abundance matrix. MEMLMe uses the assemblage embedding matrix. MEMLMc uses the abundance and the assemblage embedding matrices. Component number of WA-PLS was selected for each training set as the lowest component that showed a 5% improvement over the previous component (Table A4). WA-Cla is weighted averaging with a classical deshrinking regression, WA-Inv is weighted averaging with an inverse deshrinking regression (Birks et al. 1990). The number of WA-PLS components is selected based on the method described in 2.3.3, see Table S3 for full results.**

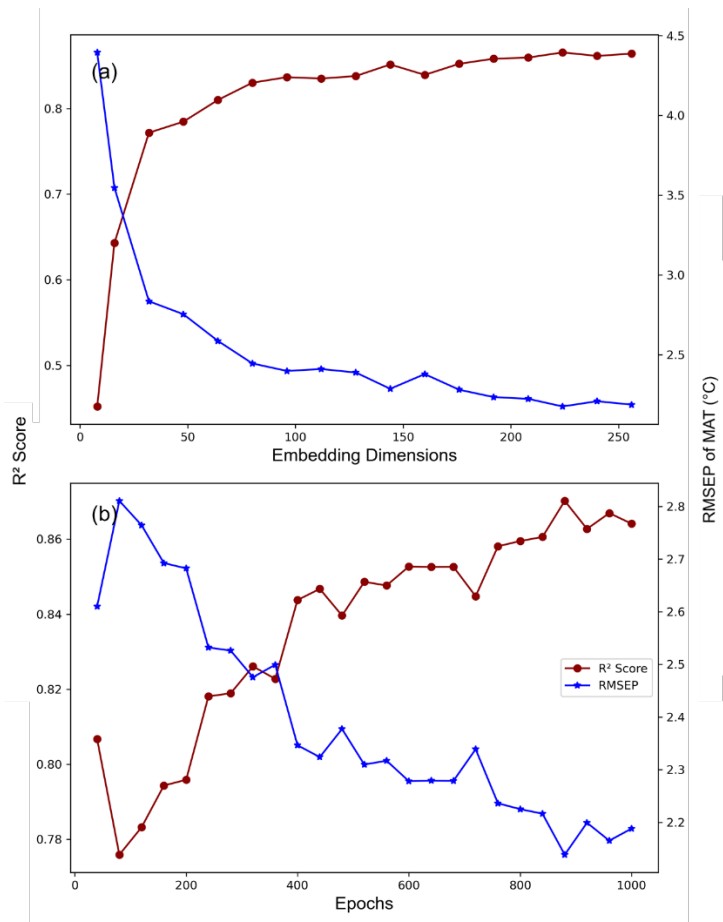

**Figure A2: MEMLMe prediction performance under different GloVe hyper-parameter settings. a) Fix epoch = 1,000, set embedding dimensions from 8 to 256; b) Fix embedding dimensions = 256, set epoch from 40 to 1,000. The model is developed from the NIMBIOS set and trained on mean annual temperature (MAT).**

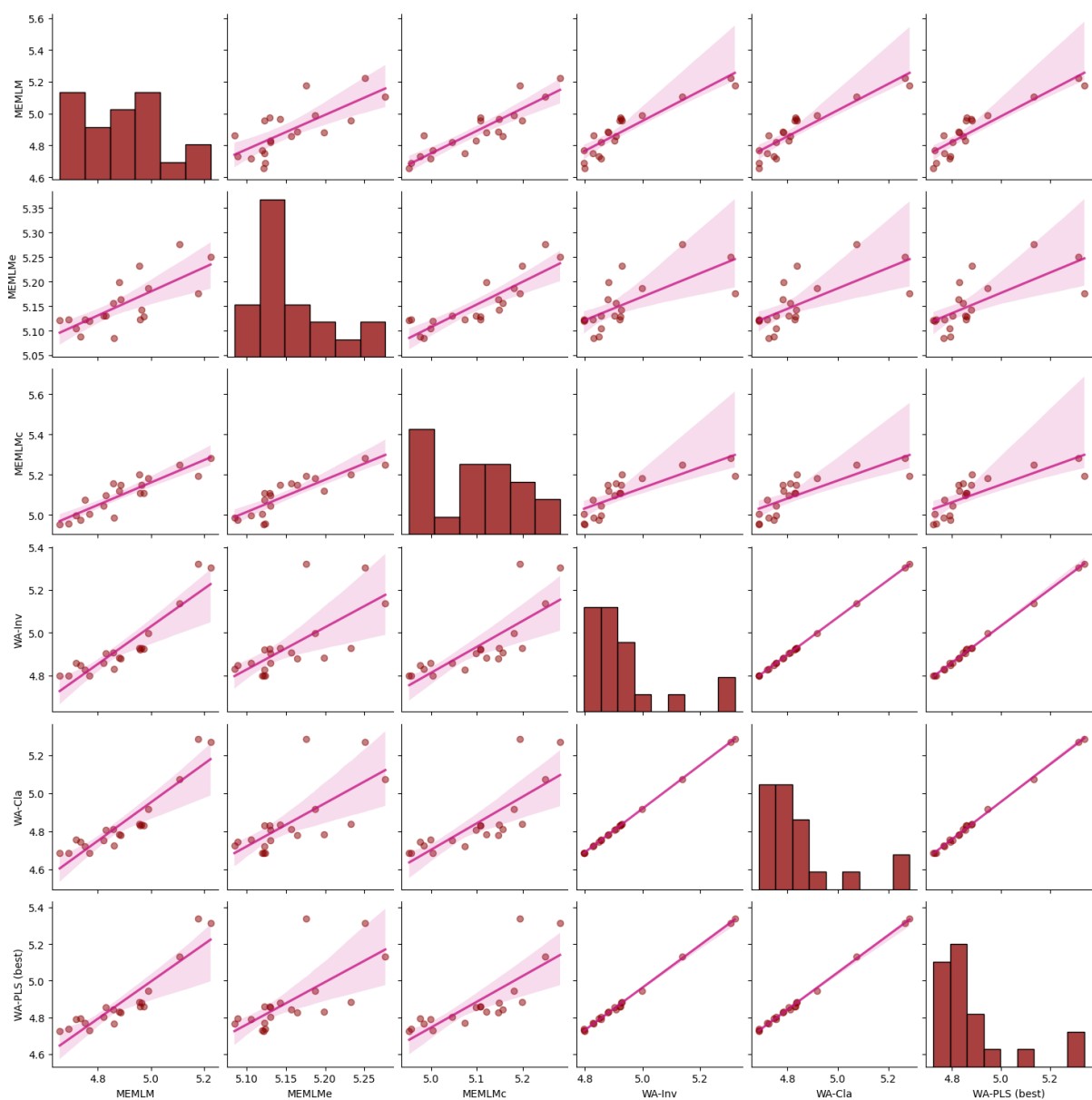

**Figure A3:** **Inter-regression of pH reconstructions for six different models for the Round Loch of Glenhead (RLGH) core.**

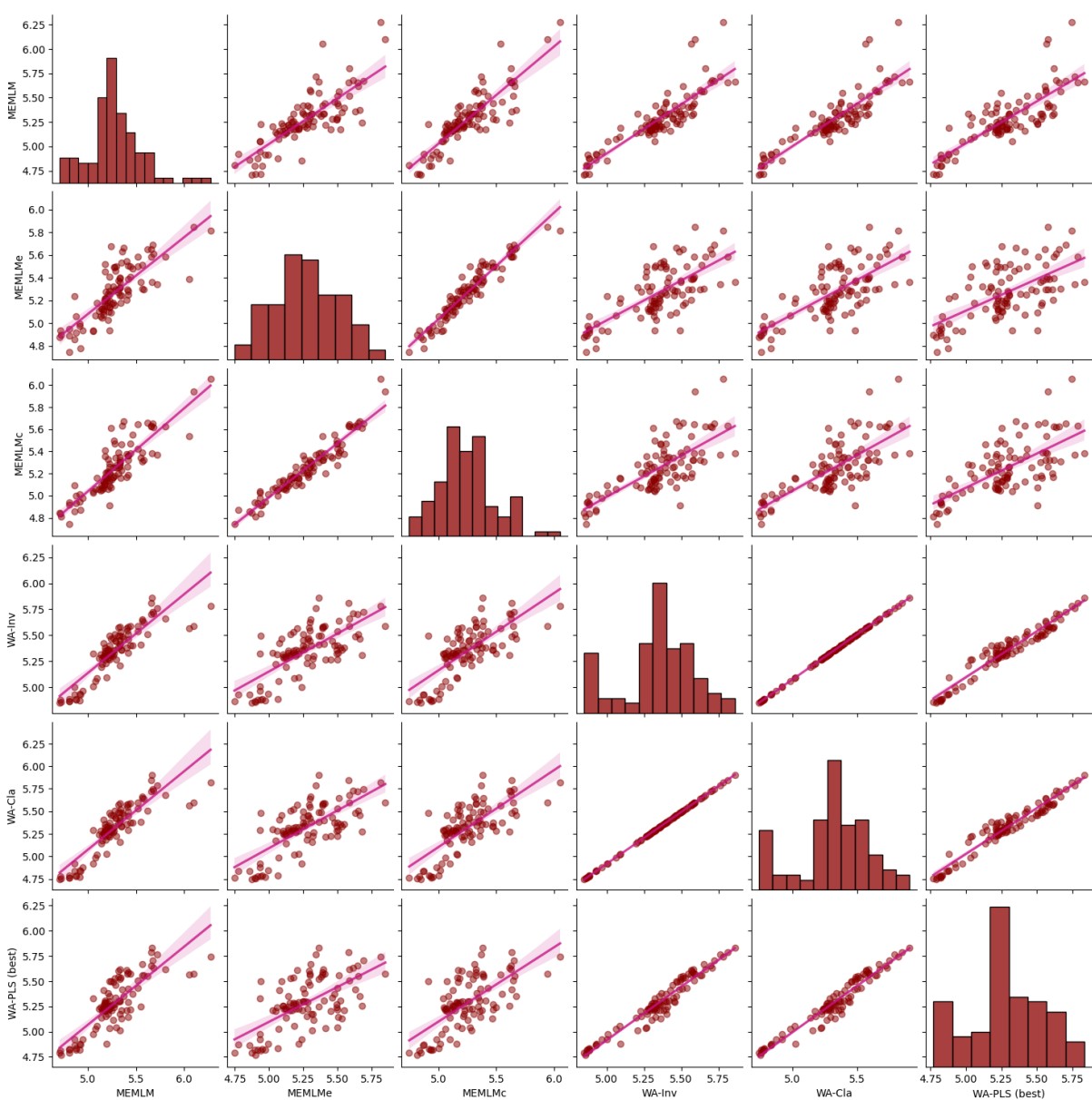

**Figure A4: Inter-regression of pH reconstructions for six different models for the Round Loch of Glenhead 3 (RLGH3) core.**

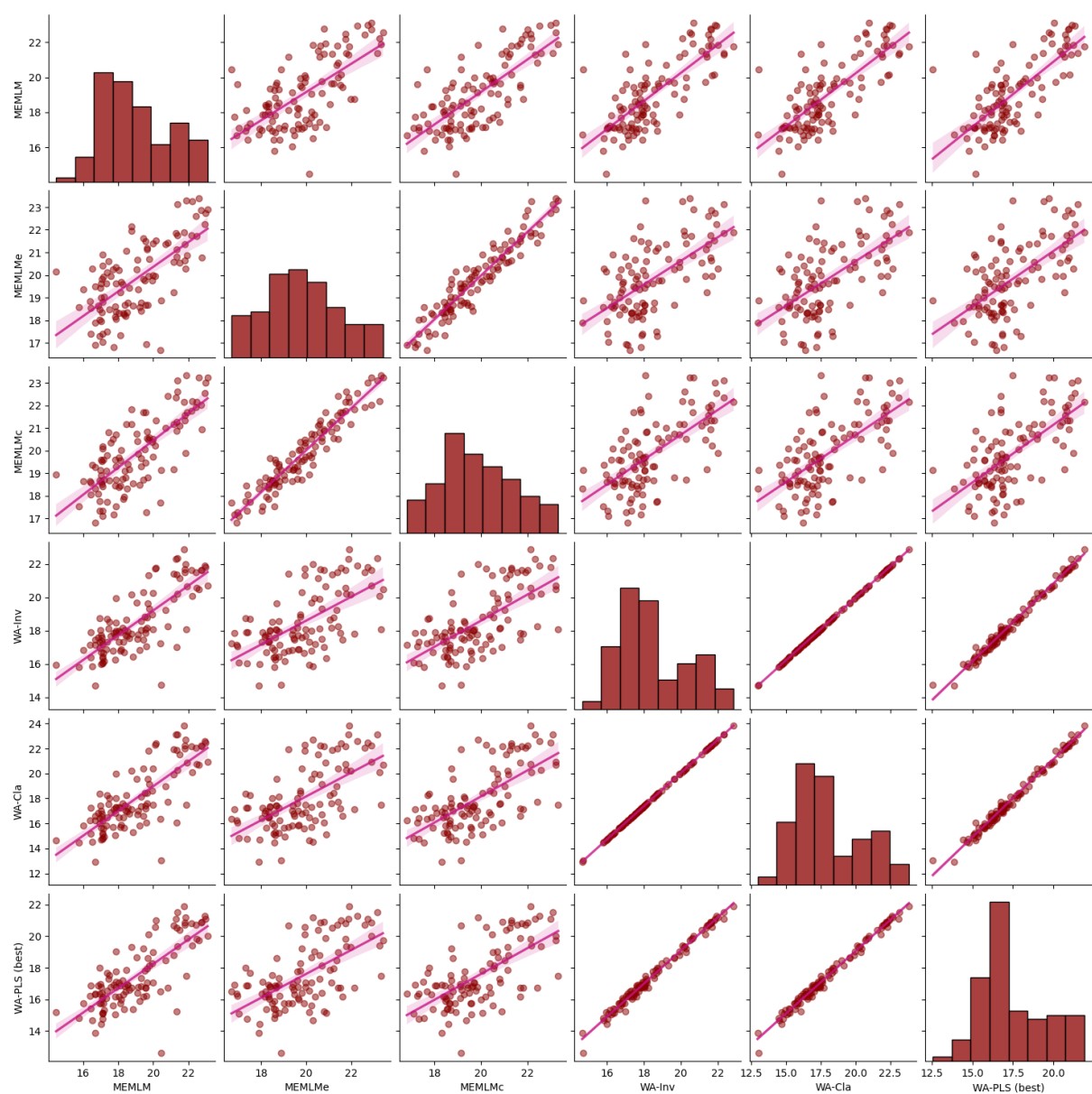

**Figure A5: Inter-regression of mean annual temperature (MAT) reconstructions for six different models for the Consuelo core.**

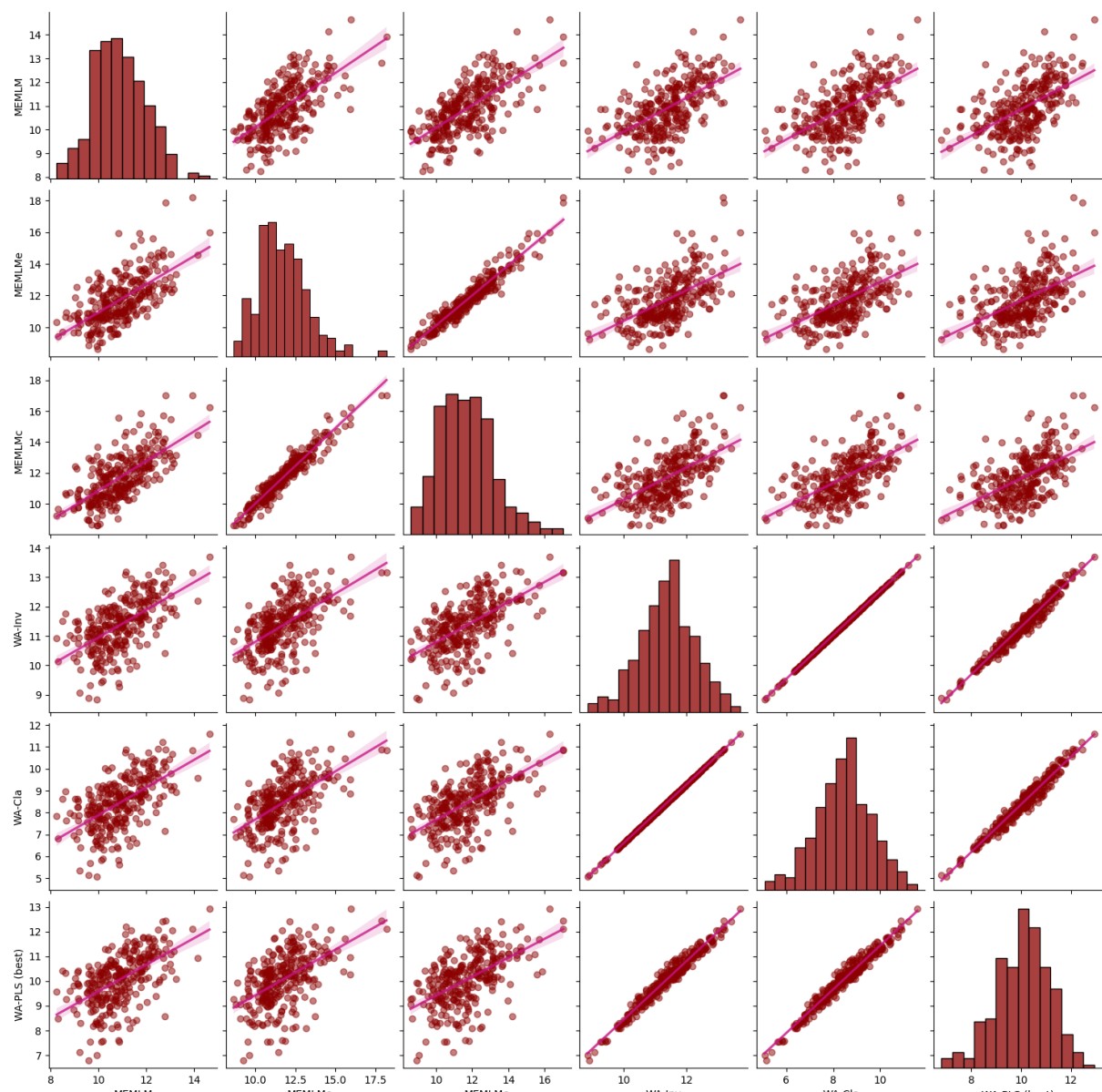

**Figure A6: Inter-regression of mean annual temperature (MAT) reconstructions for six different models for the Llaviucu core.**

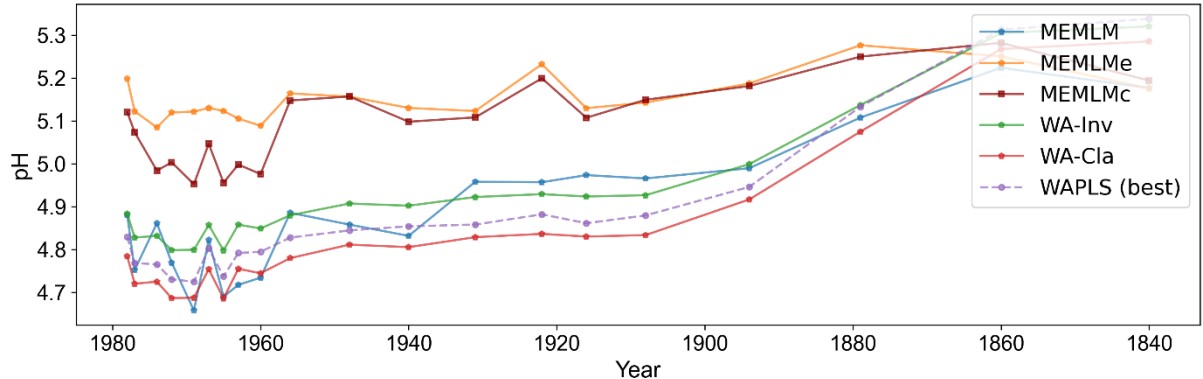

**Figure A8: pH reconstructions based on six models for the Round Loch of Glenhead (RLGH) core.**

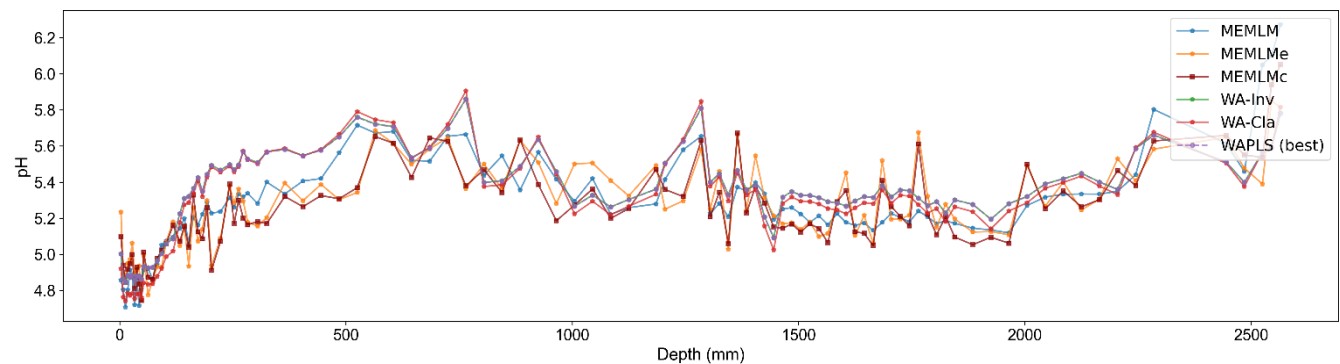

**Figure A9: pH reconstruction based on six models for the Round Loch of Glenhead 3 (RLGH3) core.**

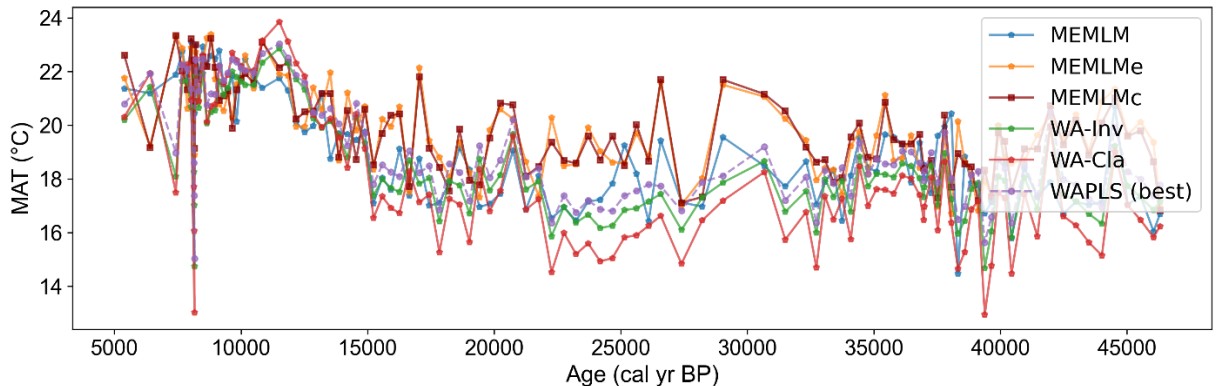

**Figure A10: Mean annual temperature (MAT) reconstruction based on six models for the Consuelo core.**

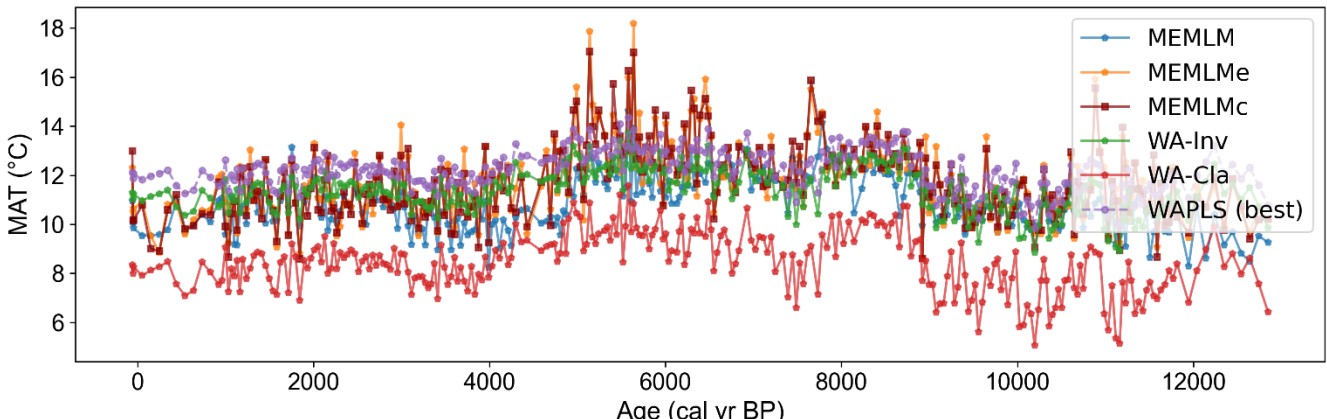

**Figure A11: Mean annual temperature (MAT) reconstruction based on six models for the Llaviucu core.**

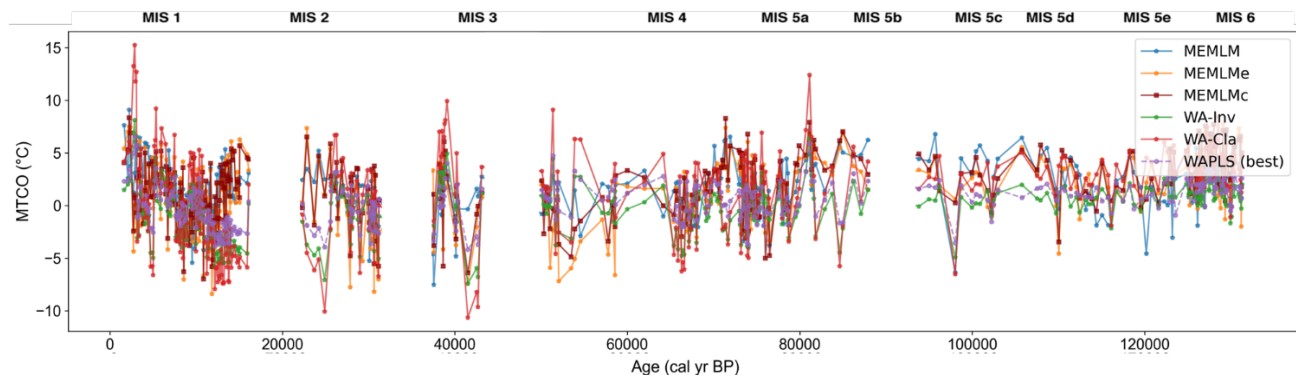

**Figure A12: Mean temperature of the coldest month (MTCO) based on six models for the Villarquemado core.**

Table A1: Root mean square error of prediction (RMSEP) and $R^2$ values (based on cross-validation) of the 18 environment elements prediction of MEMLMc in the NIMBIOS data-set. Mean is average of the prediction of the three downstream models. RF presents the random forest values; ERT presents the extra random tree results. Bold highlights the model with the best prediction performance.

|  | Elements | RF | ERT | lightGBM | MEMLMc | Mean |
|---|---|---|---|---|---|---|
| RMSEP | Precipitation of the warmest quarter | 138.17 | 131.513 | 133.124 | **125.531** | 129.042 |
|  | Isothermality | 3.065 | 2.793 | 3.09 | **2.778** | 2.838 |
|  | Annual precipitation | 483.099 | 442.623 | 479.217 | **430.291** | 445.813 |
|  | Mean temperature coldest quarter | 23.162 | 21.228 | 23.061 | **21.023** | 21.387 |
|  | Maximum temperature warmest month | 25.104 | 22.343 | 24.01 | **21.181** | 22.421 |
|  | Minimum temperature coldest month | 26.66 | 24.15 | 26.226 | **23.734** | 24.369 |
|  | Mean temperature warmest quarter | 22.898 | 21.435 | 22.655 | **20.727** | 21.316 |
|  | Precipitation of the coldest quarter | 157.458 | 135.741 | 151.69 | **129.674** | 139.075 |
|  | Precipitation of the driest month | 28.907 | 25.898 | 27.892 | **23.898** | 25.723 |
|  | Temperature seasonality | 227.1 | 203.536 | 221.23 | **203.281** | 207.248 |
|  | Precipitation of the wettest month | 64.759 | 60.669 | 64.387 | **58.822** | 60.572 |
|  | Temperature annual range | 22.179 | 20.917 | 22.101 | **20.524** | 20.83 |
|  | Mean temperature wettest quarter | 18.312 | 16.515 | 18.722 | **16.094** | 16.865 |
|  | Precipitation of the wettest quarter | 171.418 | 161.823 | 173.802 | **157.769** | 162.204 |
|  | Precipitation seasonality | 11.581 | 10.858 | 11.506 | **10.635** | 10.852 |
|  | Mean diurnal temperature range | 13.139 | 11.684 | 12.968 | **11.258** | 11.855 |
|  | Mean temperature driest quarter | 23.754 | 22.225 | 23.556 | **21.989** | 22.198 |
|  | Precipitation of the driest quarter | 96.455 | 85.831 | 92.351 | **79.657** | 85.539 |
| $R^2$ score | Precipitation of the warmest quarter | 0.656 | 0.688 | 0.68 | **0.716** | 0.7 |
|  | Isothermality | 0.862 | 0.886 | 0.86 | **0.887** | 0.882 |
|  | Annual precipitation | 0.81 | 0.841 | 0.813 | **0.85** | 0.838 |
|  | Mean temperature coldest quarter | 0.862 | 0.884 | 0.863 | **0.886** | 0.882 |
|  | Maximum temperature warmest month | 0.771 | 0.819 | 0.79 | **0.837** | 0.817 |
|  | Minimum temperature coldest month | 0.887 | 0.907 | 0.89 | **0.91** | 0.905 |
|  | Mean temperature warmest quarter | 0.85 | 0.868 | 0.853 | **0.877** | 0.87 |
|  | Precipitation of the coldest quarter | 0.845 | 0.885 | 0.856 | **0.895** | 0.879 |
|  | Precipitation of the driest month | 0.821 | 0.857 | 0.834 | **0.878** | 0.859 |
|  | Temperature seasonality | 0.835 | **0.868** | 0.844 | **0.868** | 0.863 |
|  | Precipitation of the wettest month | 0.752 | 0.782 | 0.755 | **0.795** | 0.783 |
|  | Temperature annual range | 0.848 | 0.865 | 0.85 | **0.87** | 0.866 |
|  | Mean temperature wettest quarter | 0.822 | 0.855 | 0.814 | **0.862** | 0.849 |
|  | Precipitation of the wettest quarter | 0.761 | 0.787 | 0.755 | **0.798** | 0.786 |
|  | Precipitation seasonality | 0.773 | 0.8 | 0.776 | **0.809** | 0.801 |
|  | Mean diurnal temperature range | 0.803 | 0.845 | 0.809 | **0.856** | 0.84 |
|  | Mean temperature driest quarter | 0.87 | 0.887 | 0.873 | **0.889** | 0.887 |
|  | Precipitation of the driest quarter | 0.806 | 0.846 | 0.822 | **0.868** | 0.847 |

**Table A2: Weights of the linear models in MEMLM, MEMLMe, and MEMLMc for the three training sets.**

| Weights | MEMLM | | | MEMLMe | | | MEMLMc | | |
|---|---|---|---|---|---|---|---|---|---|
| | RF | ERT | lightGBM | RF | ERT | lightGBM | RF | ERT | lightGBM |
| SWAP | -0.238 | 1.118 | 0.220 | -0.597 | 1.001 | 0.619 | -0.953 | 1.062 | 0.901 |
| NIMBIOS | -0.263 | 0.934 | 0.393 | -0.793 | 1.474 | 0.409 | -0.713 | 1.263 | 0.533 |
| SMPDSv1 | -0.106 | 0.721 | 0.431 | -0.705 | 1.180 | 0.560 | -0.340 | 0.598 | 0.773 |

RF – random forest; ERT – extra random tree; lightGBM – a gradient boosting decision tree

**Table A3: The weights of the 10 most important taxa for the environmental reconstructions in the SWAP, NIMBIOS and SMPDSv1 training sets sorted by the random forests results. Diatom taxon codes follow Stevenson et al. (1991)**

|  | Taxon | RF | ERT | LightGBM |
|---|---|---|---|---|
| SWAP | EU047A | 0.505 | 0.139 | 0.033 |
|  | AC013A | 0.072 | 0.182 | 0.028 |
|  | EU048A | 0.061 | 0.064 | 0.02 |
|  | TA003A | 0.048 | 0.043 | 0.017 |
|  | PE002A | 0.031 | 0.013 | 0.027 |
|  | CM048A | 0.023 | 0.006 | 0.029 |
|  | BR001A | 0.022 | 0.012 | 0.032 |
|  | TA004A | 0.018 | 0.02 | 0.017 |
|  | NA140A | 0.012 | 0.007 | 0.01 |
|  | CM017A | 0.011 | 0.01 | 0.019 |
| NIMBIOS | Alnus | 0.263 | 0.096 | 0.045 |
|  | Poaceae | 0.146 | 0.161 | 0.124 |
|  | Plantago | 0.118 | 0.039 | 0.006 |
|  | MoracUrtic | 0.105 | 0.02 | 0.068 |
|  | Bursera | 0.049 | 0.016 | 0.008 |
|  | Myrtaceae | 0.024 | 0.007 | 0.016 |
|  | Ericaceae | 0.022 | 0.042 | 0.021 |
|  | Hedyosmum | 0.015 | 0.03 | 0.035 |
|  | Asteraceae | 0.013 | 0.083 | 0.056 |
|  | Cyperaceae | 0.013 | 0.02 | 0.068 |
| SMPDSv1 | Picea | 0.339 | 0.038 | 0.029 |
|  | Fagus | 0.169 | 0.016 | 0.012 |
|  | Betula Chamaebetula. | 0.103 | 0.22 | 0.008 |
|  | Betula | 0.042 | 0.077 | 0.041 |
|  | Alnus Alnobetula | 0.039 | 0.017 | 0.007 |
|  | Larix | 0.03 | 0.03 | 0.009 |
|  | Quercus deciduous | 0.028 | 0.017 | 0.03 |
|  | Olea | 0.027 | 0.072 | 0.013 |
|  | Oxyria Rumex | 0.017 | 0.009 | 0.019 |
|  | Poaceae | 0.014 | 0.0144 | 0.028 |

RF – random forest; ERT – extra random tree; lightGBM – a gradient boosting decision tree

**Table A4: RMSEP (based on cross-validation) of the first five components (Comp) in weighted-averaging partial least squares (WA-PLS) of the three training sets. Bold highlights the 'best' component, noting that we accept a higher PLS component only if it exhibits a 5% improvement on the previous component (Birks 1998).**

| Data-set | Feature | WA-PLS | | | | |
|---|---|---|---|---|---|---|
| | | Comp1 | Comp2 | Comp3 | Comp4 | Comp5 |
| SWAP | pH | **0.308** | 0.299 | 0.313 | 0.327 | 0.349 |
| NIMBIOS | MAT | 5.310 | **4.979** | 4.862 | 4.840 | 4.863 |
| SMPDSv1 | MTCO | 3.207 | **2.923** | 3.022 | 3.192 | 3.365 |

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
