# Peer review of "Can machine-learning algorithms improve upon classical palaeoenvironmental reconstruction models?"

_Climate of the Past, 2023_

## Author Comment (AC1)

**Can machine learning algorithms improve upon classical palaeoenvironmental reconstruction models?**

**Peng Sun, Philip B. Holden, and H. John B. Birks**

We are grateful for these extremely thorough and constructive reviews, which have greatly strengthened our paper. Our responses are below. The reviews are in black text, our responses are in blue and edits to the manuscript in red.

**Referee #1 Cajo ter Braak**

**General:**

The purpose of the paper is unclear to me; it should perhaps be more focussed. One simple (?) purpose could be to demonstrate that particular machine learning methods can give more predictive power than classical approaches (analogue methods, WA, WA-PLS, fxTWA-PLS). Another one might be to show that combining/stacking predictors as in MEMLM improves prediction or robustness compared to using a single machine learning method (perhaps with the previous purpose included as secondary purpose).

Our primary objective is to explore whether machine-learning approaches, which do not rely upon any biological assumptions but instead have strong data-mining capabilities, especially when applied to large data sets, are capable of outperforming classical approaches. We see two sub-objectives – evaluating the ability to fit a model to a training set (and minimise RMSEP) and the ability of that model to operate under extrapolation and produce robust palaeoreconstructions.

We have added the following to the abstract.

[13] To explore the relative merits of these different approaches, we have developed a two-layered...

The current version does not give info on the second option (what is the utility of stacking) and immediately compares (three!) different MEMLM versions with WA and WA-PLS. So the possible advantages of combining the predictions of two (or three) rather similar machine learning method remains unanswered. I would be vote for adding this info to the MS.

We agree that the inclusion of stacking has slightly muddled this primary objective, as the comparison between stacked MEMLM reconstructions with unstacked classical approaches is not "apples-with-apples". The main reason we chose this strategy is because we are less interested in the relative performances of different ML algorithms, and so we simplified the comparison by presenting ML through a single consensus (stacked) reconstruction.

We have added some motivation to the introduction

[70] We build MEMLM from three different machine learning-ensemble models of random forests, extra random trees, and lightGBM. We then combine these three models into a single consensus model which we treat as our 'best' machine-learning approach.

We have quantified the benefits of stacking in section 2.1.2

"Exploratory analysis applied to the NIMBIOS data-set, building models for each of 18 environment attributes, demonstrated that the multiple linear regression approach reduced the root mean square error of prediction (RMSEP) relative to the individual reconstructions by an average of 8% (Table A1)".

The improvements in performance are clearly dominated by MEMLM itself, at least for the larger training sets where we expect ML to perform well, noting that this 8% improvement compares to the MEMLM improvements "relative to classical approaches [of] 6%... (SWAP...), 22% (NIMBIOS...) and 50% (SMPDSv1..).".

We have added the following discussion.

[137] "We note that while the stacking approach reduces RMSEP by typically 8%, we show in Section 3.1 and Table 1 that such improvements are modest relative to the improvements from the machine learning itself."

We have also added a new table A2 which isolates the performance of individual ML reconstructions (rf, etr, and lightgbm) from the stacked MEMLM reconstructions.

The reduced RMSEP using MEMLM in the bigger data sets is impressive. The different reconstructions that different versions generate signal issues. There is no attempt to provide standard errors for the (WA or consensus) reconstructions.

We have added uncertainty estimates to Table 1, calculated as the standard deviation of five randomised cross-validation exercises. Revised Table 1 and caption below.

| | | MEMLM | MEMLMe | MEMLMc | WA-Inv | WA-Cla | WA-PLS(best) |
|---|---|---|---|---|---|---|---|
| RMSEP | | | | | | | |
| SWAP | pH | **0.290 (3.7%)** | 0.331 (3.1%) | 0.296 (2.8%) | 0.308 (1.1%) | 0.317 (1.0%) | 0.308 (1.1%) |
| NIMBIOS | MAT/°C | 2.254 (1.6%) | 2.221 (1.2%) | **2.094 (1.4%)** | 3.176 (0.5%) | 3.587 (0.6%) | 2.923 (0.6%) |
| SMPDSv1 | MTCO/°C | **2.353 (0.5%)** | 2.779 (0.9%) | 2.478 (0.6%) | 5.310 (0.1%) | 6.672 (0.1%) | 4.979 (0.2%) |
| Slope | | | | | | | |
| SWAP | pH | 0.984 | 1.002 | 0.999 | 1.029 | 0.899 | 1.030 |
| NIMBIOS | MAT/°C | 0.996 | 0.998 | 0.999 | 1.005 | 0.750 | 0.996 |
| SMPDSv1 | MTCO/°C | 0.997 | 0.997 | 0.997 | 1.000 | 0.629 | 0.996 |
| $R^2$ score | | | | | | | |

| | | MEMLM | MEMLMe | MEMLMc | WA-Cla | WA-Inv | WA-PLS |
|---|---|---|---|---|---|---|---|
| SWAP | pH | **0.858** | 0.815 | 0.852 | 0.840 | 0.831 | 0.840 |
| NIMBIOS | MAT/°C | 0.856 | 0.860 | **0.876** | 0.714 | 0.635 | 0.758 |
| SMPDSv1 | MTCO/°C | **0.926** | 0.897 | 0.918 | 0.624 | 0.407 | 0.670 |

Table 1. Cross-validated root mean square error of prediction (RMSEP), regression slope, and $R^2$ score for the three training sets. All data are the means of five cross-validation exercises, which are also used to provide uncertainty estimates for RMSEP (percentage error for RMSEP in brackets). MEMLM uses the abundance matrix. MEMLMe uses the assemblage embedding matrix. MEMLMc uses the spliced abundance and embedding matrices. WA-Cla is weighted averaging with a classical deshrinking regression, WA-Inv is weighted averaging with an inverse deshrinking regession (Birks et al. 1990). WA-PLS is the 'best' model (see 2.2.3), see Table A3 for other components. Bold highlights the model with the lowest RMSEP or highest $R^2$ score.

The different versions of MEMLM (and of WA?) may generate qualitatively different reconstructions while have similar RMSEP on the training data (Fig.4 gives a nice example with same trend but different level, why?). Could other statistics of performance in current usage in WA-PLS like average bias and maximum bias and, in Liu et al 2020/2023, the regression slope, help to detect such issues? Did the different versions have different weights in their component predictors?

We have added regression slopes in Table 1 (see above)

The paper uses five-fold cross validation without further specification, so presumably using random folds. I note that cross validation gives results, such as RMSEP, that depends on the random or chosen folds, and so gives variable results on re-application. What is the error in RMSEP, so how should the reader interpret 9% improvement of one method compared to another. I note that the error is larger the smaller the number of folds.

We have added the uncertainty analysis (as above) with added text.

[239] We take RMSEP, regression slope, and $R^2$ score as performance evaluation indicators, using the scikit-learn package (Pedregosa et al., 2011). We use five-fold cross-validation. We perform each cross-validation five times with random shuffling allowing us to provide mean estimates for all validation metrics along with their standard deviations, which we provide for RMSEP.

With a larger number of folds, the error is smaller but then geographic nearness of training samples is an issue (due to spatial auto-correlation giving pseudo replication) and should be avoided, done for example in Liu et al 2020/2023.

We have added the following discussion:

[242] We note that spatial correlation and pseudo-replication within a training set can lead to overstated cross-validated performance statistics. These problems can be minimised by, for instance, removing sites that are geographically and climatically close (Liu et al, 2020, 2023). However, we include all training-set sites in cross-validation, noting that our objective is to compare the relative performances of different approaches applied to the same training sets.

The application of GloVe (in either way I think you did it, see detailed comment on L86) is similar to an old way of doing reconstruction. First apply correspondence analysis to the training data (or a principal components analysis to the training data or to the covariance matrix or innerproduct matrix obtained from the training data, which similar to the co-occurrence matrix used in GloVe ) and then use the dimensions, i.e. the sample scores of the various axes, in a second step (in this paper called (integrated) embedding vectors), which used subsequently in a multiple linear regression step (similar to the random forest in the paper). See Roux 1979 cited in (ter Braak & van Dam 1989). If the full dimensionality, is used (instead of using only the first few dimensions), this type of approach should not be worse than one using the training data directly, as the embedding is/should contain a full representation of the training data. In summary: the GloVe approach does an unconstrained ordination of the training assemblage data (i.e. without environment data) and uses the dimensions (ordination axes / embedding vectors) so obtained for supervised learning of environment as a function of ordination axes. In my view, this approach is not likely to be superior so more simple direct approaches.

I note that the GloVe model is log-linear with free row and column parameters, so that is an RC-model in the sense of Goodman ((Goodman 1981; Ihm & van Groenewoud 1984; Goodman 1986; Goodman 1991)). Correspondence analysis is close to a log-linear model (Supporting Information to (ter Braak & te Beest 2022)) and both models are close to the Gaussian model (with equal tolerances) that was the basic motivation for WA and WA-PLS.

Indeed, there are similarities with approaches that use ordination to dimensionally reduce taxa matrices before building the transfer function. We further agree regarding the cost function, that GloVe is a logarithmic linear model. However, there are some notable differences. GloVe places emphasis on encoding taxa rather than efficient dimensionality compression. GloVe does not enforce orthogonality and therefore preserves the semantic properties of taxa. In GloVe, each dimension is relatively equal, rather than hierarchically explaining information from high to low.

We therefore expect that GloVe offers more interpretability than traditional dimensionality reduction methods, as it looks for dimensions which imply similar meaning. We therefore anticipate that GloVe encoding should directly represent ecological niche information, and its ecological niche information is derived from species co-occurrences rather than species abundance-environmental values. Although our analysis has not convincingly demonstrated reconstruction improvements from the inclusion of GloVe, it has demonstrated that GloVe is able to successfully encode

assemblage relationships. We hope this will justify its potential for further investigation in future work.

We have added the following clarifying text.

[173] We note that while the GloVe algorithm closely resembles unconstrained ordination, GloVe emphasises semantics, seeking dimensions which convey meaning and which have relatively similar importance. This contrasts with unconstrained ordination, which focuses on the explanation of variance and dimension ordering. By focusing on semantics, we expect that GloVe will provide more interpretability than traditional dimensionality reduction methods.

GloVe uses the log(co-occurrences) for good reasons. Did you consider in the other approaches transformations of the assemblage taxon proportions? Could that not be handled naturally in your multiple ensemble approach?

Following Pennington et al. (2014), the use of log(co-occurrences) is deemed reasonable. We agree other transformations would be interesting to consider, but we feel these are beyond the scope and ambitions of the paper.

Consider adding some motivation why you do not consider modern-analogue methods or, alternatively, limit your conclusions.

We have now clarified throughout the manuscript that we specifically consider WA approaches and have added the following to the conclusions.

[389] We have focussed only on a comparison with WA approaches, which are the most widely used reconstruction technique, being simple to apply, well understood, and straightforward to interpret.

The Discussion also needs attention. I found a number of remarks that appear to lack (precise) support. Some of these may only be answered using simulated data as in (ter Braak *et al.* 1993).

**Details:**

*Abstract*

You use twice constucts like "three models of A, B and C". I vote for "three models (A, B and C)".

Thank you, we have made this change in both places.

L19 "MEMLMe, which uses embedded assemblage information" The intended audience is not likely to know what embedded assemblage information means. Even a google search would not be of much help, I believe. Something with dimension reduction or unconstrained ordination of co-occurrence information might be more helpful.

Rephrased to

"MEMLMe, which uses only dimensionally reduced assemblage information…"

L20 "MEMLMc which incorporates both taxon abundance and assemblage data." What is the difference between taxon abundance data and assemblage data? Unclear.

We have clarified this to say

"incorporates both raw taxon abundances and dimensionally reduced summary (GloVe) data."

L24 "embedded assemblage information" See L19.

Changed to "dimensionally reduced (GloVe) data"

L25 Why "However"?

We have deleted "however"

L26 Make more clear that the different version of MEMLM also generated qualitatively different reconstructions.

We have rephrased this to

[26] When applied to fossil data, MEMLM variants sometimes generated qualitatively different palaeoenvironmental reconstructions from each other and from reconstructions based on WA approaches.

L26 Here you switch from present to past tense. Put all in past tense.

Done

L28 "catastrophically fail" Where did you find this and why this emotional term?

We have toned down the language to "fail badly".

The clearest example is perhaps the MEMLMc reconstruction of Llaviucu where "All three methods [MEMLM, MEMLMc and WAPLS] display similar overall trends with mid-Holocene warming, but each display different centennial variability, which for the MEMLMc reconstruction is clearly unrealistic for the Holocene, with temperature excursions as large as 8°C."

L50: should -> could

Done

L51 Delete "However,"

Done

L51-54 Optimal methods and algorithms can be derived by making assumptions. The GloVe paper by Pennington et al gives a nice example! But, algorithms themselves never make assumptions (except for data properties, e.g. they may fail to work on negative data or symbolic data). A particular algorithm is only motivated from/ derived from assumptions. For example, taking the mean does not assume a sample from a normal distribution (even for P/A data and Poissonian counts the mean is optimal!), but it is a fine summary of the location of data when such assumptions hold true. WA has been derived from the unimodal model for the ecological niches of taxa (with equal niche breadths), but it does not assume such models; it may well work well for other (strictly compositional) data, but one does not know whether it is best in some sense. Even modern analogue methods are 'based on' assumptions. The assumptions lead to the choice of a proper/the best measure of distance or similarity between the fossil sample and the training samples.

Thank you, we have rephrased to "based upon"

[53] Machine-learning approaches are not based upon any biological assumptions, which may weaken their performance relative to mathematically simpler classical approaches that are. For instance, weighted averaging (WA) approaches are based upon the simple but informative assumption that taxa have a unimodal response to the environmental variable of interest (ter Braak and Barendregt, 1986).

L56-7 These questions could already be addressed by any one of the three machine learning methods. It would be of interest what benefit there is of the superlearner/stack model  approach in MEMLM compared to using a single ensemble method only.

Please see earlier discussion in response to this point.

And, could WA or WA-PLS contribute to the multiple ensemble approach?

Yes, we agree they might be the case. However, we feel this is beyond the scope of this study, noting that our primary motivation for stacking was not to analyse the benefits of stacking per se, but rather to present the MEMLM approach through a single consensus reconstruction that was not dependent upon a specific ML algorithm. See earlier in our response.

L57 "to apply in" -> "and carried out"

Done

L59: "The benefit of machine learning is that it has strong data mining and information extraction ability. An associated problem," With ability, data mining and information extraction being vague terms, I find this a sentence without meaning. With "associated

problem" in the next sentence, the writers appear to me to agree that the previous sentence is problem.

We have clarified the meaning with this rephrasing:

[61] The benefit of machine learning lies in its robust data mining and information extraction capabilities, especially when applied to large data-sets. Data mining involves discovering patterns, trends, and correlations hidden within extensive data-sets. Information extraction, on the other hand, focuses on extracting insights from unstructured data, typically relying on Natural Language Processing and encoding techniques to understand and analyse semantic information embedded within unstructured data. An associated problem is that when a sample size is limited, machine learning is more likely to learn the noise component and generate prediction errors due to over-fitting...

L63 & L140: The reference to the fourth-corner paper (Legendre et al. 1997) appears ill-chosen to me. The paper is cited for "reduce [] over-fitting errors" and "co-existence & and environment", while it is a paper that focusses on traits in **behavioural ecology, it is not even on environment.**

We have removed this citation.

L64: "This is the motivation for the ensemble learning approach we present, namely the Multi Ensemble Machine Learning Model (MEMLM)" It is unclear to me from this sentence whether MEMLM is a new method developed and explained in this paper or an existing method. No reference is given so it looks like new/novel although it does not sound new to me. Please cite earlier work in this direction. I know for example the "super learner" approach by van der Laan (https://doi.org/10.2202/1544-6115.1309) and google gave me (Naimi & Balzer 2018).

Indeed, ensemble learning is not novel. We have clarified this by adding

[68] Ensemble learning was developed to address these issues (Zhou, 2012).

L65: "Classical studies [] reconstruction approaches" cites. Norberg et al. 2019 but this paper may contain "integret[ion]", but is not on reconstruction of palaeoenvironment.

We apologise for this error and thank you for picking it up. Norberg et al 2019 is a classic study on species distribution models.

We replace "reconstruction approach" with "ecological approach."

L66: "does not attribute weights to" -> "gives equal weight to"

Done

L67-72: There is an issue with this bit on what is a 'weight'. I note that a linear multiple regression model has regression coefficients that are often referred as "weights". A predictor (taxon) with a small regression coefficient has small weight, and a predictor with a large (absolute) coefficient has a large weight, a large influence whereas the predictors are initially unweighted (no user-defined weights). In this sense even WA and WA-PLS weigh taxa differently in their transfer function. In TWA-PLS there is indeed an additional weight called the tolerance. The relevant description is mainly derived from Liu et al.'s discussions.

We have rephrased as

[70] We build MEMLM from three different machine learning ensemble models of random forests, extra random trees, and lightGBM. We then combine these three models into a single consensus model which we treat as our 'best' machine learning approach. Classical studies have integrated different ecological approaches by calculating the mean of their predictions (Norberg et al., 2019). An arithmetic mean gives equal weight to each model, even though the models may have different advantages in different applications (Schulte and Hinckley 1985; Zhou 2012). In MEMLM, we weight each model according to its predictive power under cross-validation.

Most classical models give equal weight to different taxa, which may reduce their prediction potential and smooth the reconstruction (e.g. Brooks and Birks, 2001; Heiri et al., 2003; Battarbee et al., 2005; Wei et al., 2021a). In WA-PLS (TWA-PLS), tolerance down-weighting can be applied to assign weights to each taxon in reconstructing the environment that depends upon the breadth of the taxon's environmental niche (Liu et al., 2020). Bayesian approaches such as BUMPER (Holden et al., 2017) are built on classical assumptions and are highly constrained by taxa with low environmental tolerances, especially when characterised with high confidence. In machine learning ensemble models, each taxon has a different predicted contribution which is used to weight its contribution to the ensemble.

L74: "In MEMLM, we apply both taxon weights and model weights." So I hoped that under Methods these weights would be clearly introduced or explained, but I did not find such explanation. Also, I would expect in Results the weight given to each model.

The explanation of taxon weighting is found in 2.3.2 (now retitled "The prediction importance indicator for taxon weighting"), Table A4 (now retitled weights of the "10 most important taxa"). We have also added Table A2, which provides model weights.

| weights | MEMLM | | | MEMLMe | | | MEMLMc | | |
|---------|-------|-----|---------|-------|-----|---------|--------|-------|---------|
| | rf | ert | lightgbm | rf | ert | lightgbm | rf | ert | lightgbm |
| SWAP | -0.238 | 1.118 | 0.220 | -0.384 | 0.901 | 0.536 | -0.605 | 0.856 | 0.790 |
| NIMBIOS | -0.263 | 0.934 | 0.393 | -0.793 | 1.474 | 0.409 | -0.713 | 1.263 | 0.533 |
| SMPDSv1 | -0.106 | 0.721 | 0.431 | -0.533 | 1.041 | 0.518 | -0.492 | 0.777 | 0.739 |

Rf – random forest; ert – extra random tree; lightgbm – a gradient boosting decision tree

Table A2: Weights of the linear models in MEMLM, MEMLMe, and MEMLMc for the three training sets.

L74: Why do you use in MEMLM two, or even three, very similar methods. All are based on decision trees and are already ensemble methods by their own. However, they are based on different algorithms and implementation approaches, implying that their advantages may vary with different datasets.

We do agree that it is the case that their advantages may vary with different data-sets. According to the 'No Free Lunch' principle, it is unlikely for a single model to be the best on all tasks. Our primary motivation to integrate them is to provide a single consensus model and minimise concerns that would arise using specific models, with the additional expectation of achieving relatively better predictive performance across different sample data-sets. Our final stacking weight is adaptive based on the training data of the data-set, rather than artificially fixed. See responses above on this point.

L79: "includes" -> uses. And add the type of encoding/mention GloVe

Revised to

[83] We develop three versions of MEMLM; the standard version which only considers raw taxon abundance data; MEMLMe, which only uses dimensionally reduced (GloVe) assemblage data; and MEMLMc, which uses both.

L81: The abbreviation NLP is not/rarely used later on, so delete.

Done

L84: The term "environmental assemblages" is unknown to me. Change.

Revised to "taxon assemblages"

L86: You write "GLOVE  to generate the embedding vectors of different taxa in different samples based on assemblage information" suggesting that GloVe  is applied to the assemblage information, whereas on L84 you write "In environmental assemblages, there are analogous co-occurrence relationships between taxa which we hypothesize convey information on their ecological functioning" which suggest to me that GloVe  is applied to a (perhaps weighted) co-occurrence matrix.

The second approach gives vectors for taxa (their meaning in your language analogy), as mentioned explicitly on L86. But: you do not describe how those vectors were transferred to vectors for samples for use in Random forest and the like (Layer one of figure 1)), except for the phrase "and then to integrate the embeddings within each sample to represent the assemblage", but you do not describe how you did this, except that you write on L159 "the assemblages can be described as linear combinations of the

features", but you do not describe how (I guess: as in principal components analysis but more detail is needed, likely in an appendix). And whatever how you did this, describe why that is a good way (or the best way) to do it. If you followed the first approach, you applied the log-bilinear model (or RC (Goodman's row-column) model) to the primary training data and did not calculate a co-occurrence matrix.

Note that co-occurrence is a thing that is usually calculated from 1/0 (binary) primary data whereas you have compositional data. So this need explanation/details as well.

We have clarified these point as follows:

[89] In taxon assemblages, there are analogous co-occurrence relationships between taxa which we hypothesise convey information on their ecological functioning. We therefore use GloVe to generate embedding vectors by considering the frequency of co-occurring taxon pairs across the training set. We then concatenate the embedding vectors of each sample to represent the assemblage.

L91: classical reconstruction approaches -> WA and WA-PLS as there are more approaches than WA around, notably modern analogue methods.

We have replaced "classical reconstruction approaches" with "classical WA-based reconstruction approaches"

L130-131. Note that this uses the full data set. So the resulting RMSEP and R2 are not crossvalidation RMSEP and R2 in a formal sense. This should be remarked in the Discussion

All of the performance statistics throughout the paper use cross-validation, and RMSEP is correct in the formal sense. We emphasise this with the following edit

[137] A consensus reconstruction based on the mean of the three ensemble approaches also improved predictive power but reduced cross-validated RMSEP errors relative to the individual reconstructions by an average of 5%.

L146-159 The Pennington et al paper is a great paper in my view. Whereas all you write here is in the paper to motivate the approach does not understandably summarize GloVe to me. GloVe is a row-column bilinear model of the form (r_i + c_k + R_i*C_j) fitted to the log-transformed co-occurrence matrix derived from the primary data. It is fitted by weighted least-squares to log(co-occurrence count +1) [so as to avoid problems with log 0]. GloVe is thereby very close to unconstrained ordination models used in ecology except for the transformation to co-occurrences , see Suppl Info in (ter Braak & te Beest 2022) and discussion in (ter Braak 1988).

Thank you for this clear description of the mathematical formulation of GloVe. Our aim was to provide text to make the approach accessible to a broad audience, but we have added this very useful mathematical summary.

[144] In formal terms, GloVe is a row-column bilinear model of the form (r_i + c_k + R_i*C_j), least-squared fitted to the log-transformed co-occurrence matrix derived from the primary data. GloVe is thus very close to unconstrained ordination models used in ecology except for the transformation to co-occurrences (ter Braak and te Beest, 2022, ter Braak, 1988).

L159-160: So in essence the same assemblage data is entered twice (original and transformed) as predictors in Layer one (figure 1). Apparently, the authors have little confidence that ensembles of  decision trees can figure out all interesting combination of taxa…. (and perhaps they are right, except that the result of MEMLM and MEMLMc look rather similar). Actually, all differences between the MEMLM versions would show that basic machine learning methods used in layer one 'fail' in one way or another..

Yes, that is the motivation, we have added the following at the start of section 2.1 MEMLM

[102] There are three variants of MEMLM, with the scientific motivation to explore i) whether decision trees can extract all useful information (MEMLM), or, if not, ii) whether GloVe can improve this (MEMLMc) and iii) whether GloVe alone is sufficient to encode assemblage data (MEMLMe).

L147: I would call it a conditional probability. [but this text should be removed/replaced anyway, see before].

We have changed to "conditional probability". We prefer to keep this paragraph (augmented by the mathematical summary), which we believe will help less mathematically expert readers understand the underlying motivation and philosophy of the approach.

**Section 2.2 Use the same order everywhere  (in 2.2.1 and 2.2.2) from smallest to largest training data set : SWAP, NIMBIOS, SMPDSv1.**

We follow this order (SWAP, NIMBIOS, SMPDSv1) throughout.

L198. Model parameters are not "Performance and validation metrics".

We have modified the section title to "Model parameters, performance, and validation metrics".

L199 under -> using

Done

L199 frame -> Python software library.

To further enhance clarity we have added

[216] PyTorch provides a set of tools and interfaces to implement, train, and deploy deep-learning models.

L199-200 Delete "for … feedback"

Done

L209 "explore to integrate" Rephrase.

Rephrased to

[227] explores different ways to integrate each taxon's abundances to increase predictive power.

L201 256 dimensions: for all data sets? With this large number of dimensions, the GloVe solution (if properly done) should almost be identical to the (possibly transformed) training assemblages.

Yes, we agree this should be the case. The focus of GloVe is on extracting semantic meaning. In linguistics, typically 200 dimensions of meaning are sufficient to encode a language and our baseline analysis does not assume the approach will be more efficient for an assemblage decomposition. We address this in section 3.1 and associated figure A2.

"We performed additional cross-validation tests on MEMLMe to confirm that the embedding approach does indeed encode useful information, noting that with an embedding dimension of 256 (comparable to the number of taxa in the training sets) we are not applying the approach under significant dimensional reduction. We applied a progressively increasing embedding dimension applied to an MEMLMe model of MAT using the 533-taxon NIMBIOS data-set (Figure A2). This sensitivity demonstrates that only about 30-dimensions are required for MEMLMe to outperform WA-PLS (RMSEP 2.914°C), so that that dimension reduction by more than an order of magnitude retains sufficient information to build a useful model. Increasing the embedding dimension towards 256 unsurprisingly progressively improves RMSEP further by encoding additional assemblage information."

L204 Which Python? function/file in https://github.com/Schimasuperbra/MEMLM is the rewritten function, so that we can check it?

The name of that package is 'GloVe-python'. Unfortunately, it is no longer maintained and we have rewritten it to ensure that it is available. All code will be made available and documented.

L213 "upstream model" First usage of this term. Meaning?

Rephrased to "for each of the three machine-learning models".

L218. See earlier comment on cross-validation and its dependence on the folds or randomness.

See earlier response in L130-131.

L223 "tests" ?-> guards against?

Done

L233 scikit-learn package. What role has this package? Is it a Python package?

We have clarified

[259] scikit-learn package, which is a powerful machine learning Python package which incorporates the most widely used machine-learning algorithms and related data processing and validation functions.

L240 "The additional learning power with increasing training-set size is evident." Say simpler.

Rephrased to

"The benefits of machine-learning approaches clearly increase with increasing train-set size."

L244 "MEMLMe consistently under- performs relative to MEMLM and MEMLMc" Say simpler.

Rephrased to

"However, MEMLMe is consistently the worst performing MEMLM variant (albeit generally better than the WA approaches), and so we do not use it in the reconstructions.

L249 FiguA2a

Done

L248-249 Rephrase. How does this fig look like for the much small SWAP data set?

Our objective is to illustrate that the embedding approach is able to encode useful information, which we achieve by showing that the most taxon-rich assemblage (NIMBIOS, with 533 taxa) can be usefully represented with only 30 dimensions. We have removed the generalised conclusion, rephrasing to

[278] We performed additional cross-validation tests on MEMLMe to confirm that the embedding approach can encode useful information, noting that with an embedding

dimension of 256 (comparable to the number of taxa in the training sets) we are not applying the approach under significant dimensional reduction. To explore this, we applied a range of embedding dimensions to the MEMLMe model of the richest data-set, being the 533-taxon NIMBIOS data-set (Figure A2a). This sensitivity demonstrates that 30 dimensions are sufficient for MEMLMe to outperform WA-PLS (RMSEP 2.914°C) in this training set. Figure A2b illustrates the learning power of increased training, with RMSEP increasing by around 0.4°C as the number of training epochs is reduced from the 1,000 we used to 40.

L250 Rephrase.

Done (above)

L253 "unsurprisingly" ? What about the danger of overfitting?

Good point, we have deleted "unsurprisingly"

L254 Change to: as the number of training epochs is decreased from the 1000 we used to 40.

Done

Table 1 Legend. Make sure we know for sure the R2 is also cross-validated and does not only apply to the RMSEP (where the last P is, to me, already suggesting prediction under cross-validation).

We have added to the caption

All data are the means of five cross-validation exercises, which are also used to provide uncertainty estimates for RMSEP (percentage error for RMSEP in brackets).

And near the top of section 2.3.4

[240] We perform each cross-validation five times with random shuffling allowing us to provide mean estimates for all validation metrics along with their standard deviations, which we provide for RMSEP.

L268 "lowest RMSEP" but two MEMLM models in the figs. I would vote for all three MEMLM models, WA-Cla and WA-PLS (at least in a supplement).

All models are now plotted in Appendix figures A8 to A12, and introduced in 3.2 (together with scatterplots, requested below)

[305] In the Appendix, Figures A3 to A7 illustrate scatterplot matrices of all six reconstruction approaches, and Figures A8 to A12 compare reconstructions for all six models through time.

[Figure]

Figure A3: Inter-regression of pH reconstructions for six different models for the RLGH core.

[Figure]

Figure A4: Inter-regression of pH reconstructions for six different models for the RLGH3 core.

[Figure]

Figure A5: Inter-regression of MAT reconstructions for six different models for the Consuelo core.

[Figure]

Figure A6: Inter-regression of MAT reconstructions for six different models for the Llaviucu core.

[Figure]

Figure A7: Inter-regression of MTCO reconstructions for six different models for the Villarquemado core.

[Figure]

Figure A8: pH reconstructions based on six models for the RLGH core.

[Figure]

Figure A9: Mean annual temperature (MAT) reconstruction based on six models for the RLGH3 core.

[Figure]

Figure A10: Mean annual temperature (MAT) reconstruction based on six models for the Consuelo core.

[Figure]

Figure A11: Mean annual temperature (MAT) reconstruction based on six models for the Llaviucu core.

[Figure]

Figure A12: Mean temperature of the coldest month (MTCO) reconstruction based on six models for the Villarquemado core.

L269 reverse WA-PLS and "the best classical approach"

Done

L270 PLS component 2 -> PLS using 2 components

Done

L276 Rephrase avoiding "understates". Say simpler.

Rephrased to "but with reduced acidification"

L280 Refer to Figure 2 earlier in the text.

Done

Figure 6a. Please add another figure (in Suppl?) with a scatterplot matrix of the 3 (or 4) reconstructions (against one another). Similarity and dissimilarity are hard to see in Fig.6a, but may be present.

Done, see above

L317 inconsistent. In which sense? Rephrase.

Changed to "incoherent"

Results section:

No info is given on the weights of the machine learning approaches in the consensus nor on their individual (truly) crossvalidatory RMSEP and R2.

See previous responses. We re-emphasise that all summary statistics throughout the paper were generated under cross-validation.

Discussion:

L325-327 As you had 3 machine learning methods in your MEMLM and three version of MEMLM, it is easy for a reader to get confused. I certainly was on first reading. I missed for example on first reading that the lines in the figs are already consensus reconstructions. But not that all methods could be included...

We have added text to emphasise this at the start of section 2.1

[101] There are three model variants (MEMLM, MEMLMe, and MEMLMc), each of which takes different inputs (Figure 1), which is the only difference in their construction. The scientific motivation for the three variants is to explore i) whether machine learning decision trees can extract all useful information (MEMLM), or, if not, ii) whether GloVe can improve this (MEMLMc), and iii) whether GloVe alone is sufficient to encode assemblage data (MEMLMe). Each variant is built using the same three machine-learning approaches (random forests, extra random trees, and lightGBM), which are combined into a single consensus reconstruction model for each.

L327 Is multiple regression a weak learner approach? In which sense? Simply: delete "a weak learner approach based on".

Deleted

334-336 "We note that the real power of embedding (dimension reduction) approaches in ecology is likely to be in their applications to much larger data-sets, when ecological relationships between 10,000's of taxa and their environment are being considered." Likely? 1) the sentence is unclear in the sense whether it is the number of samples or the number of taxa that is "larger". If the number of taxa is large or huge (compared to the sample size?), dimension reduction might help by reducing the number of predictors. Dimension reduction might help to reduce 'noise'.

We may be missing your point, but we believe this text is unambiguous, "when ecological relationships between 10,000s of taxa and their environment are being considered"

L349-342 Say simpler. I do not believe that we want "a more complete description of a data-set". We want robust prediction. "suggesting" why not "we show on three data sets that …."

We have rephrased to

[393] These improvements in performance clearly validate the potential benefits of strong data-mining abilities of machine learning, suggesting these techniques have the potential to improve upon classical reconstruction approaches.

L357-358. A "However" linking two different methods? Intentional?

These two approaches both use embedding, which we have made clearer

[420] The additional complexities of incorporating embedding information in MEMLMc does not reduce RMSEP or spurious variability and neither does it improve statistical significance. However, MEMLMe demonstrates that embedding is useful as it can summarise ecological assemblages using significantly fewer dimensions.

L359 "felt"? See at L334-336.

Changed "felt more clearly" to "clearer"

L362 "Even though all models were applied under the same extrapolation, the WA-PLS2 reconstructions were found to be more reliable than MEMLM, although WA-PLS2 also failed to generate robust reconstructions at Villarquemado." 1) Two although's in the same sentence. The first should not be a "though": OK, all models used the same data, and had thus to face the same issue with potential extrapolation (say, where in particular). 2) What is the evidence in your results or in the literature for the remark "the WA-PLS2 reconstructions were found to be more reliable than MEMLM"?

Rewritten to

[402] Both MEMLM and MEMLMc approaches fail on the Llavuicu core, confirming our suspicion that the unrealistic variability is an artefact even though the overall trends of the reconstruction are consistent with the robust WA-PLS2 reconstruction. All three approaches fail the statistical robustness test at Villarquemado, which is sensitive to multiple environmental factors and has responses which appear too complex to be captured by a single explanatory variable.

L356 "ensure the reconstructions". Now it reads as if significance testing can help making the reconstructions more robust. It can only help avoiding accepting/publishing non-robust reconstructions.

Changed "ensure" to "confirm"

L375 "available on request". This is very old-fashioned. Make them available with credits where credit is due.

To our knowledge, this particular set of data has not been made public and is not our data to share. Mark B. Bush would be willing to assist in providing the data.

L380 I would like to see more code for making the paper reproducible by somebody knowledgeable in [both R and] Python. Add the type of software : R, Python, script or function.

We have now made the entire data utilisation process and methods available on GitHub.

Figure A1 legend. "Regression visualization" -> "Scatter plots"

Done

Table A2 What do the values represent. How are they defined? What parameter in the output is it? In which sense are they useful to the reader?

See earlier response

Cajo ter Braak Wageningen Oct 20. 2023.

Goodman, L.A. (1981) Association models and canonical correlation in the analysis of cross-classifications having ordered categories. *Journal of the American Statistical Association,* **76,** 320-334

Goodman, L.A. (1986) Some useful extensions of the usual correspondence analysis approach and the usual log-linear models approach in the analysis of contingency tables. *International Statistical Review,* **54,** 243-270.https://doi.org/10.2307/1403053

Goodman, L.A. (1991) Measures, models and graphical displays in the analysis of cross-classified data. *JASA,* **86,** 1085-1138

Ihm, P. & van Groenewoud, H. (1984) Correspondence analysis and Gaussian ordination. *Compstat Lectures,* **3,** 5-60

Naimi, A.I. & Balzer, L.B. (2018) Stacked generalization: an introduction to super learning. *European Journal of Epidemiology,* **33,** 459-464.https://doi.org/10.1007/s10654-018-0390-z

ter Braak, C.J.F. (1988) Partial canonical correspondence analysis. *Classification and related methods of data analysis* (ed. H.H. Bock), pp. 551-558. Elsevier Science Publishers B.V. (North-Holland) http://edepot.wur.nl/241165, Amsterdam.http://edepot.wur.nl/241165

ter Braak, C.J.F., Juggins, S., Birks, H.J.B. & Van der Voet, H. (1993) Weighted averaging partial least squares regression (WA-PLS): definition and comparison with other methods for species-environment calibration. *Multivariate Environmental Statistics* (eds G.P. Patil & C.R. Rao), pp. 525-560. North-Holland, Amsterdam.https://edepot.wur.nl/249353

ter Braak, C.J.F. & te Beest, D.E. (2022) Testing environmental effects on taxonomic composition with canonical correspondence analysis: alternative permutation tests are not equal. *Environmental and Ecological Statistics,* **29,** 849-868.https://doi.org/10.1007/s10651-022-00545-4

ter Braak, C.J.F. & van Dam, H. (1989) Inferring pH from diatoms: a comparison of old and new calibration methods. *Hydrobiologia,* **178,** 209-223.http://dx.doi.org/10.1007/BF00006028

**Referee #2 Andrew Parnell**

The paper by Sun et al is an enjoyable paper to read about the use of some newer machine learning (ML) methods applied to palaeo-environmental reconstruction. The ML approaches are well described and the approach is mostly easy to follow. Overall I found it a bit disappointing that the authors did not take a probabilistic ML approach to the problem given that uncertainty quantification is such an important part of reconstruction. We are left with an approach that seems to simply provide a best estimate of climate change over time. A bootstrapping or Bayesian ML extension of this work would have been most welcome to better compare the approaches.

Thank you for this suggestion. We have now added uncertainty quantification, described in new section 2.3.3

[234] Uncertainty quantification is provided for all machine-learning reconstructions using IBM's UQ360 uncertainty quantification package. We apply the Infinitesimal Jackknife, which performs a first-order Taylor series expansion around the maximum likelihood estimate, to provide 25% and 75% confidence intervals of the prediction (IBM, 2024)

I was excited and then quite confused about how the GLOVE model was used in the process. I had never come across GLOVE before and am still unclear as to why it was chosen as an embedding method. My understanding is that these approaches are somewhat similar to PLS in that they can reduce the dimensions of the inputs in a clever way so as to capture the majority of the information. Though unlike PLS the dimension reduction seems not to be based on the response variable. I'm not really sure why it is required in this approach as none of the data sets would be considered particularly high dimensional, nor did computation seem like a barrier to performance. The GLOVE approach in particular seemed like an odd choice since it seems to throw away the data values in lieu of presence/absence (I might be reading this wrong). If so, perhaps an autoencoder or a modern dimension reduction approach such as UMAP might have been more appropriate? The argument seems to be that the GLOVE model will capture interactions between proxy values, but it is unclear to me why the tree-based ML models aren't doing this already. Perhaps the lack of benefit of using GLOVE, as measured by RMSE and R2, is because it's not an entirely suitable approach. I certainly feel that a clearer explanation of why GLOVE is used would be helpful.

Our motivation for GloVe is interpretability of the embedding. There are strong similarities with approaches that use ordination to dimensionally reduce taxa matrices before building the transfer function. However, there are some notable differences. GloVe places emphasis on encoding taxa rather than efficient dimensionality compression. GloVe does not enforce orthogonality and therefore preserves the semantic properties of taxa. In Glove, each dimension is relatively equal, rather than hierarchically explaining information from high to low.

We therefore expect that GloVe offers more interpretability than traditional dimensionality reduction methods, as it looks for dimensions which imply similar

meaning. We therefore anticipate that GloVe encoding should directly represent ecological niche information, and its ecological niche information is derived from species co-occurrence rather than species abundance-environmental values. Although our analysis has not convincingly demonstrated reconstruction improvements from the inclusion of GloVe, it has demonstrated that Glove is able to successfully encode assemblage relationships. We hope this will justify its potential for further investigation in future work.

We have added the following clarifying text in section 2.1.3.

[173] We note that while the GloVe algorithm closely resembles unconstrained ordination, GloVe emphasises semantics, seeking dimensions which convey meaning and which have relatively similar importance. This contrasts with unconstrained ordination, which focuses on explanation of variance and dimension ordering. By focusing on semantics, we expect that GloVe will provide more interpretability than traditional dimensionality reduction methods.

Finally, in the conclusions I think it's really important to point out that these models have some fundamental flaws which make them not really suitable for widespread use just yet. The most obvious one to me is the lack of uncertainty quantification mentioned above, but another is the lack of a time series model being included. The autocorrelation in both the training sets and the fossil reconstruction period is usually considerable, and as this quite often changes between the calibration and fossil periods, is a really important aspect of the models.

We now address the uncertainty quantification as detailed above.

As far as we are aware, there have been no attempts at allowing for temporal autocorrelation in the fossil data and for spatial autocorrelation in the calibration data in palaeoreconstructions. Such a problem is beyond the scope of this study and would form the basis of a new and demanding study. We have added the suggested caution

[402] We note that autocorrelation in both the training sets and the fossil reconstruction period is usually considerable, and this often changes between the calibration and fossil periods.

Some less important points:

- Introduction: the term 'space instead time' seems grammatically incorrect and not widely used (as far as Google tells me).

We have rephrased this to:

[37] By considering environmental variability across space instead of through time...

- Section 2: **The first paragraph repeats the last paragraph above**

Thank you, we have deleted the first instance.

- Section 2.51. It's slightly confusing to state that at the embedding dimension was set to 256 when later in the paper it's shown that you only need ~30. Again, it's not clear whether the GLOVE approach is being used here to capture interactions (beyond what trees will capture?) or to reduce the dimension of the problem, in which case 256 seems like overkill. A little bit more discussion would be helpful.

We have added the following in the discussion

[377] Our motivation for retaining 256 embedding dimensions in MEMLMe is that the focus of GloVe is on extracting semantic meaning. In linguistics, typically 200 dimensions of meaning are needed to fully encode a language. While we have shown that far fewer dimensions are sufficient to build a good reconstruction model, demonstrating the explanatory power of the most important embedding dimensions, there are progressive improvements in performance as dimensional size increases. This demonstrates that less important dimensions provide useful explanatory information, and potentially additional understanding and interpretability.

- Section 2.4. It would be nice to have some kind of computational speed estimates for running this models beyond just stating the hardware.

We have clarified

[263] The computational time taken for 5-fold cross-validation of the MEMLMc model is 138 seconds (SWAP), 406 seconds (NIMBIOS), and 2834 seconds (SMPDsV1).

- Figure 2 (and similar figures) it's not clear to me why the histograms for WA-PLS are so different from the others. As someone who doesn't use frequentist techniques I'd like a little more explanation of what's happening here.

Thank you for this question. To our knowledge the shapes of these histograms has not been discussed in the literature. We have added some text in the discussion (paragraph 6) based on our tentative ideas about histogram shape. What seems clear is that machine-learning approaches generally fail badly when trained with randomised data (the histograms are left-skewed and most instances explain very little down-core variance). In contrast, WA approaches tend to explain a substantial portion of down-core variance even when randomised, which may reflect their correlative nature, so that reconstructions are dominated by a few abundant species. We have added the following

[407] The shapes of the histograms of the proportion of variance explained in the RLGH and RLGH3 pH reconstructions based on diatom data and randomised modern SWAP training pH values in the significance testing are very different for WA-PLS1 and for MEMLM and MEMLMc (Figs. 2, 3). Such differences contrast with the more consistent histogram shape for the significance-test results for the other sequences where the reconstructions are based on pollen data (Figs. 4–6). Machine-learning approaches generally fail badly when trained with randomised environmental data as the

histograms are left-skewed and explain little down-core variance (Figs. 2–6). In contrast, the WA-PLS1 pH reconstructions (Figs. 2, 3) based on diatom data explain a substantial amount of the down-core variance even when the modern pH data are randomised (Figs. 2, 3). This may result from the short and dominant environmental gradient in the SWAP diatom–pH training data and the high inherent correlation and dominance of a relatively few abundant taxa within the modern and fossil diatom data. The pollen training data, however, used for the MAT or MTCO reconstructions of the other sequences (Figs. 4–6) are large (638 and 6,458 samples) and hence cover longer and more complex environmental gradients than the pH training data (167 samples). It is also likely that the pollen data, both modern and fossil, are influenced by multiple environmental factors, not only MAT or MTCO.

- Code availability. Please mention this in the introduction or abstract. The code is well commented, but it's a shame there isn't more data or instructions as to how to reproduce the results.

We have included additional comments in the model and provided download links for each data-set, along with data citation information. Moreover, we have added a manual document describing the whole data-processing procedure.

- Table A1. As there's a regression model here I was hoping to see something about the weights (and their uncertainties) on the different models. Do you really need all three or are some models strongly preferred over others for different data sets?

The weights are now provided in Table A2, reproduced below.

| weights | MEMLM RF | ERT | lightGBM | MEMLMe RF | ERT | lightGBM | MEMLMc RF | ERT | lightGBM |
|---|---|---|---|---|---|---|---|---|---|
| SWAP | -0.238 | 1.118 | 0.220 | -0.597 | 1.001 | 0.619 | -0.953 | 1.062 | 0.901 |
| NIMBIOS | -0.263 | 0.934 | 0.393 | -0.793 | 1.474 | 0.409 | -0.713 | 1.263 | 0.533 |
| SMPDSv1 | -0.106 | 0.721 | 0.431 | -0.705 | 1.180 | 0.560 | -0.340 | 0.598 | 0.773 |

---

## Author Response (AR2)

**Public justification (visible to the public if the article is accepted and published)**:
Dear authors,
as you can see from the second round of reviews (in fact just one review), some unresolved issues remain and need to be addressed. In particular, the reviewer indicates that contrary to his earlier suggestions, the article has not become more focused. Also, his point (3) from the earlier review has not been addressed. I would like to point out that I will not send this article for another round of reviews but will scrutinize the revised version for the exact implementation of the reviewer's remaining concerns. Also, the reviewer has graded "presentation quality" as "fair" (2 out of 4). I would like you to improve it towards publication.
I look forward to reading the revised version and remain at your disposal for any queries.
Kind regards,
Irina

Dear Irina,
With regard to presentation, we have improved the histrogram plots in Figures 2 to 6. With regard to focus, we have added the following paragraph at the end of the introduction. In our view, each of the components listed by Cajo ter Braak are needed to address the question whether machine learning can improve upon classical techniques.

In summary, there are several aspects to the question of whether machine-learning algorithms can improve upon classical reconstruction methods. Our strategy to address these has three components

1) There are many ensemble machine learning algorithms, and there is no reason to prefer any of these a priori. To address this, we apply three widely used approaches of random forests, extra random trees, and lightGBM. We combine these into a single consensus reconstruction to simplify comparisons and provide the 'best possible' reconstruction.

2) Natural language-processing models are a widely used dimensional reduction approaches in machine learning, and we apply one such method, GloVE, to supplement ensemble machine learning trained on raw count data. We explore whether this approach can usefully encode assemblage information to either i) improve the reconstructions based only on raw count data - unlikely given that dimension reduction does not provide additional information, but not ruling out the possibility that data transformation can assist the learning or ii) replace the raw count data, increasing numerical efficiency and potentially providing information on ecological functioning.

3) It is not sufficient that a reconstruction approach performs well on a training set. It must also be statistically robust when applied to independent core data, which likely lies outside the high-dimensional space of the training set. We cannot assume that machine learning and classical approaches perform equally well under extrapolation. Therefore, we do not only apply conventional tests of cross-validated RMSEP, regression slope and R2, derived solely from the training set, but we also consider the statistical significance of core reconstructions, applying the technique of Telfrod and Birks (2011)

**Review of revision 1 of cp-2023-69 : "Can machine-learning algorithms improve upon classical**
**palaeoenvironmental reconstruction models?**
**Peng Sun, Philip. B. Holden, and H. John B. Birks"**

General:

The revision as mostly additions compared to the original one. This paper is in fact four papers in one: (1) a comparison of machine learning (decision tree based) methods and weighted averaging methods for paleo reconstruction (2) the use of multiple ensemble methods to lever the performance of individual methods (3) the idea of using embedding by the GloVe method, that is, re-expressing the taxon and co-occurrence data (in a possibly lower number of dimension) and using the re-expressed data either alone or together with the original data as predictors in the machine learning methods (4) showing the good cross-validatory performance in terms of RMSEP does not guarantee a reliable reconstruction. To identify un- or less- reliable reconstructions the authors recommend the Telford/Birks significance test. This start of my review is a rephrase of the first one in my review of the first version, which started: "The purpose of the paper is unclear to me; it should perhaps be more focussed.". The authors did not make the paper more focussed.

I have few or no comments on the paper regarding points (1) and (4). My reservation in the original submission on point (2) has been resolved (I understand the authors claim in the rebuttal by now, see below). I see many issues with point (3) which is perhaps the most novel to paleo-ecology but also the least important.

With regard to focus, we have added the following paragraph at the end of the introduction. In our view, each of the components listed by Cajo ter Braak are needed to address the question whether machine learning can improve upon classical techniques.

In summary, there are several aspects to the question of whether machine-learning algorithms can improve upon classical reconstruction methods. Our strategy to address these has three components

1) There are many ensemble machine learning algorithms, and there is no reason to prefer any of these a priori. To address this, we apply three widely used approaches of random forests, extra random trees, and lightGBM. We combine these into a single consensus reconstruction to simplify comparisons and provide the 'best possible' reconstruction.

2) Natural language-processing models are a widely used dimensional reduction approaches in machine learning, and we apply one such method, GloVE, to supplement ensemble machine learning trained on raw count data. We explore whether this approach can usefully encode assemblage information to either i) improve the reconstructions based only on raw count data - unlikely given that dimension reduction does not provide additional information, but not ruling out the possibility that data transformation can

assist the learning or ii) replace the raw count data, increasing numerical efficiency and potentially providing information on ecological functioning.

3) It is not sufficient that a reconstruction approach performs well on a training set. It must also be statistically robust when applied to independent core data, which likely lies outside the high-dimensional space of the training set. We cannot assume that machine learning and classical approaches perform equally well under extrapolation. Therefore, we do not only apply conventional tests of cross-validated RMSEP, regression slope and R2, derived solely from the training set, but we also consider the statistical significance of core reconstructions, applying the technique of Telford and Birks (2011)

GloVe is an unconstrained ordination (dimension reduction) method applied to the pairwise taxon co-occurrence table in the training set. The main text says that the GloVe scores (in MEMLMc) are appended to the taxon abundance values (and used directly in MEMLMe). On this second/third reading I missed how scores for training samples are derived. This key information is kind of 'hidden' in line 170 which has the issue that it uses the term assemblage data in at least two ways: (1) co-occurrence matrix (2) training samples containing taxon percentages. Clarify.

We added the following after the first sentence of section 2.2 Assemblage Data:

"The SMPDSV1 (Harrison, 2019) and SWAP (Stevenson et al., 1991) datasets record the percentage of each taxon in each sample, whereas the NIMBIOS dataset uses integer counts.  When constructing the co-occurrence matrix, whether the data are integer counts or percentages, we sum that data during co-occurrence.""

As an aside and simple analogy: a principal components analysis can be carried out on the covariance matrix and a non-centred one on the inner-product matrix, which is very close to a co-occurrence matrix when applied to presence/absence data [similar things apply to correspondence analysis, which presumably comes even closer a co-occurrence matrix]. This is known as R-mode PCA. From R-mode PCA, the usual sample scores can be derived by taking a linear combination (section 5.3.6 of (Jongman, ter Braak & van Tongeren 1995). From this analogy it can be conjectured that analysing co-occurrence gives very little (probably, nothing) extra compared to analysing the abundance matrix itself. A way to find out is described by (van der Voet 1994). It would be nice (but not a prerequisite), in my view, to add such analysis to the MS.

We would prefer not to add further complications to the paper.

From a theoretical point of view do not think an ordination analysis of co-occurrences can really improve paleo-reconstruction or significantly lower RMSEP. The reason is that decision-tree based methods combine the predictors themselves. Such combinations are interactions in terms of classical statistical models and have co-occurrence as special case.

We have added a caveat that improvements are unlikely (and indeed we found no improvement). However, applying transformations to input data, can help statistical models to learn and we are not convinced that potential improvements should be ruled out a priori. We have added the text:

"We explore whether this approach can usefully encode assemblage information to either i) improve the reconstructions based only on raw count data - unlikely given that dimension reduction does not provide additional information, but not ruling out the possibility that data transformation can assist the learning or ii) replace the raw count data, increasing numerical efficiency and potentially providing information on ecological functioning."

It is unclear to me from the text how the co-occurrence matrix has been calculated as each sample contains taxon percentages. So I do not know whether the co-occurrence value of taxa j and k in a sample is calculated from the taxon percentages or from taxon presence/absence in a sample. In the latter case the maximum number of co-occurrences is the number of samples in the training set

See above response re section 2.2.

Details:

L32 I would like to have this conclusion to be separate from the comparison which is the main focus of the paper. I suggest to add "also" to the sentence or, in full, "Apart from the comparison between machine learning and weighted averaging method for paleo reconstruction we also conclude ..."

We prefer to connect these statements rather than to separate them i.e.

"Given these conclusions, we consider that..."

L24 "embedded assemblage data" first occurrence of embedding. I suggest to change the sentence to "the three MEMLM approaches performed... as judged by cross-validatory prediction error in the larger training data sets.

We replace "embedded" by "dimensionally reduced" and "under cross-validation" with "as judged by cross-validatory prediction error".

L29 "could fail badly" add : in the reconstruction??

Done

L33 "cross-validation" Change in line with the line 24 change.

We prefer to leave this unchanged as this statement applies to any metric derived under cross-validation, not only the prediction error.

L61-63 The text, as I read it, suggests that "data mining" and "information extraction" are used here in the meaning of "supervised" and "unsupervised" learning. I wonder whether information extraction is not a misnomer (even if usual in the ML word). What about using the new term "representation learning" for unconstrained ordination/factor analysis?

We prefer not to create new terminology, noting that we specifically define what we intend these terms as meaning.

L64 I do not think that the phrase "understand and analyse semantic information" makes sense. Semantics is about meaning, so that an aim can be to 'extract/obtain semantic information by an analysis'. Please rephase and avoid the usage of the term semantic in ecological context as it is unclear what is supposed to mean (i.e. avoid terms that sound impressive but do not carry meaning for an ecologist).

We have changed "semantic information" to "relationships".

L84-86 I do not know what are "dimensionally reduced (GloVe) assemble data" and what is "the more complex versions" [yes, the ones using GloVe, which has not yet been introduced]. Rephase.

Done

Figure 1. Is it really impossible to change Raw num to Row num in the fig.?

Done

L113 "develop the assemblage matrix" To me, the assemblage matrix is the same as the abundance matrix, which makes the sentence strange. Rephrase.

Done

L131 It should be said explicitly that the multiple regression is applied to each of the five folds, so as to enable calculation of the cross-validatory prediction error (RMEP) without further analyses (I missed this/did not think about it in this way in the first version). Also, add that for any down-core application of the model, the model is recomputed for all data. And as all has been done five times, the total number of analyses is 5 (folds) x 5 (replications of the cross validation) + 1 (for the final model used for reconstruction). Is this the correct interpretation of what has been done?

We clarify with the following text:

"The three upstream models are applied to reconstruct the training data-set and we then build a multiple linear regression model to fit these reconstructed values to the actual value in the training set. To fit the multiple linear regression model, we apply internal 5-fold cross-validation for each model separately and use the predictions from this cross-validation to fit regression weights. We then treat the consensus model as a

single encapsulated model and perform 5-fold cross-validation, each time using 80% of the training set. The total validation computation therefore comprises five internal cross-validations and one regression fit."

L137 "the stacking approach" First appearance of stacking. Might be unclear. Rephrase or explain.

Done

L139-140 Table A2 could be supplemented with the standard errors (or percentage error, if defined explicitly) of the coefficients based on the five folds (the root mean variance across the five replications).

The weight table is based on the entire training set and the linear model obtained from this training are used to reconstruct paleoclimate, so the coefficients uniquely define our model. While we could provide uncertainties associated with the model's construction, we feel this is diverging further from our motivation for stacking, which is only to provide our 'best' possible ML approach for fair comparison with classical approaches.

L145. You give the one-dimensional form of the model here (copying from my first review). Either mention this explicitly, or extend to $R_i^{\wedge'} C_j$. A point that I did not find in Pennington et al 2014 is that a co-occurrence matrix is a symmetric matrix (although they describe/word it asymmetrically as "$X_{ij}$ tabulate[s] the number of times a word j occurs in the context of word i") so that R and C in the formula should be identical, shouldn't they? So my question is: is your co-occurrence matrix symmetric? And what about the numerical values that you obtained? And, are the sample scores then linear combinations of the R or of the C value (if different).

Since the GloVe model uses the gradient descent algorithm to iteratively minimize the error, the embedding (R) and (C) will not be identical, but they are at least very close.

L145. "least-squared fitted" -> fitted by weighed least-squares"

Done

L146 "except"-> "except, perhaps," See my notes under General.

Done

L156 Here is the place to describe how you calculated co-occurrence from percentage abundance data.

See response above

At line 158, we have replaced "assemblages" with "co-occurrence matrix"

L157 Delete "The objective … functioning." as it carries little information relevant to paleo-reconstruction.

We prefer to keep this. It does carry little direct relevance to paleo-reconstruction, but it has the potential to aid understanding of the reconstruction and was one of our motivations for applying the approach.

L170 This key sentence should be rephrased (see under General).

See above.

L170-171 Move to L147.

Done

L148. Add, for example, "It may be helpful to describe the motivation for this particular row-column model." [Lines 148-169 describe motivation for the row-column model; this is how Pennington et al. came up with this row-column model. Note that there are older but similar ways to motivate this model; it is particular attractive for strictly compositional data].

Done

L173-177. I read here: GloVe dimensions emphasises meaning. Really? In my view, there is no contrast with unconstrained ordination. Please, delete.

Deleted

L233-235 The infinitesimal jackknife requires a twice differential model (Extrinsic UQ Algorithms — uq360 0.1 documentation). Are decision tree models twice differential? (I presume not). A topic for future research is to try and validate this approach. (Birks et al. 1990) used a bootstrap approach.

Sorry for the description error and we thank ter Braak for pointing out the problem. Yes, infinitesimal jackknife is unsuitable for non-differentiable decision trees. By looking at the source code of UQ360, we found that its external uncertainty method is meta-model, which uses another decision tree model to predict the error of the basic model. We have changed the description to:

"UQ360 utilizes meta-models to estimate the uncertainty bounds of the preserved models, providing upper and lower limits on prediction errors. Specifically, it employs additional decision tree models to capture and re-estimate the prediction errors of the source models."

L240 each [five-fold] cross-validation?

We have changed "Cross-validation" to "Each five-fold cross-validation".

L255 Specify which variance. Now I have to reread Telford and Birks to find out. They write "proportion of variance in the fossil data explained by a single reconstruction" "estimated using" "redundancy analysis". Add this info.

Done

L282. "This sensitivity" Make more precise.

Done

L 293 "percentage error" When I google percentage error I obtained a formula for a single estimate compared to the true value, e.g. Percent Error: Definition, Formula & Examples - Statistics By Jim. You have more values. Please define more precisely or give a reference that contains explicitly and clearly precisely what you used.

Done

L294 "spliced abundance and embedding matrices." Spliced? Embedding matrix is not easy to understand either.

We have changed spliced to combined. Embedding is well defined e.g. section 2.1.3

Figure 2 and similar. "b, c & d) statistical significance" I see histograms and within it a p-value. Rephrase.

Changed to statistical significance testing.

L402 encode Both?

fixed

L404. WA-PLS2 does not occur in the first part of the sentence. So why "though"? Rephrase.

We have deleted this sentence

L401 "fail badly" In which sense?

This is explained in the sentence, they fail because the histrograms are left skewed and explain little down-core variance.

L401-2 shorten to: "Machine learning approaches trained with randomised environmental data yield left-skewed histograms, showing they explain little down-core variance as is natural (desired?) for randomized environmental data (Figs. 2–6)".

L444 Add the GitHub location or (preferred) give it a zenodo DOI, so that all is reproducible in principle.

We have uploaded the code to Zenodo, https://zenodo.org/records/13138593

Figure A1 I would say "y vs x" as vertical vs horizontal (ordinate vs abscissa)
Fig and legend are inconsistent in this sense.

We have changed to "observed values vs  predicted values"

Cajo ter Braak Wageningen July 1. 2024

Birks, H.J.B., Line, J.M., Juggins, S., Stevenson, A.C. & ter Braak, C.J.F. (1990) Diatoms and pH reconstruction. Philosophical Transactions of the Royal Society London, Series B, 327, 263-278.https://doi.org/10.1098/rstb.1990.0062
Jongman, R.H.G., ter Braak, C.J.F. & van Tongeren, O.F.R. (1995) Data analysis in community and landscape ecology. Cambridge University Press, Cambridge.0-521-47574-0
van der Voet, H. (1994) Comparing the predictive accuracy of models using a simple randomization test. Chemometrics and Intelligent Laboratory Systems, 25, 313-323